# The AAA-ATPase Yta4/ATAD1 interacts with the mitochondrial divisome to inhibit mitochondrial fission

Jiajia He[1,2,☯], Ke Liu[1,2,☯], Yifan Wu[1], Chenhui Zhao[1], Shuaijie Yan[1], Jia-Hui Chen[1,3], Lizhu Hu[4], Dongmei Wang[1], Fan Zheng[1], Wenfan Wei[1], Chao Xu[1,2], Chengdong Huang[1,2], Xing Liu[1,2], Xuebiao Yao[1,2], Lijun Ding[5], Zhiyou Fang[4]*, Ai-Hui Tang[1,3]*, Chuanhai Fu[1,2]*

1 MOE Key Laboratory for Cellular Dynamics & Center for Advanced Interdisciplinary Science and Biomedicine of IHM, Division of Life Sciences and Medicine, University of Science and Technology of China, Hefei, China, 2 Anhui Key Laboratory of Cellular Dynamics and Chemical Biology & Hefei National Research Center for Interdisciplinary Sciences at the Microscale, School of Life Sciences, University of Science and Technology of China, Hefei, China, 3 Institute of Artificial Intelligence, Hefei Comprehensive National Science Center, Hefei, China, 4 Anhui Province Key Laboratory of Medical Physics and Technology, Institute of Health and Medical Technology, Hefei Institutes of Physical Science, Chinese Academy of Sciences, Hefei, China, 5 Center for Reproductive Medicine, Department of Obstetrics and Gynecology, The Affiliated Drum Tower Hospital of Nanjing University Medical School, Nanjing, China

☯ These authors contributed equally to this work.
* z.fang@cmpt.ac.cn (ZF); tangah@ustc.edu.cn (A-HT); chuanhai@ustc.edu.cn (CF)

**Data Availability Statement:** All relevant data are within the paper and its Supporting Information files.

## Abstract

Mitochondria are in a constant balance of fusion and fission. Excessive fission or deficient fusion leads to mitochondrial fragmentation, causing mitochondrial dysfunction and physiological disorders. How the cell prevents excessive fission of mitochondria is not well understood. Here, we report that the fission yeast AAA-ATPase Yta4, which is the homolog of budding yeast Msp1 responsible for clearing mistargeted tail-anchored (TA) proteins on mitochondria, plays a critical role in preventing excessive mitochondrial fission. The absence of Yta4 leads to mild mitochondrial fragmentation in a Dnm1-dependent manner but severe mitochondrial fragmentation upon induction of mitochondrial depolarization. Overexpression of Yta4 delocalizes the receptor proteins of Dnm1, i.e., Fis1 (a TA protein) and Mdv1 (the bridging protein between Fis1 and Dnm1), from mitochondria and reduces the localization of Dnm1 to mitochondria. The effect of Yta4 overexpression on Fis1 and Mdv1, but not Dnm1, depends on the ATPase and translocase activities of Yta4. Moreover, Yta4 interacts with Dnm1, Mdv1, and Fis1. In addition, Yta4 competes with Dnm1 for binding Mdv1 and decreases the affinity of Dnm1 for GTP and inhibits Dnm1 assembly in vitro. These findings suggest a model, in which Yta4 inhibits mitochondrial fission by inhibiting the function of the mitochondrial divisome composed of Fis1, Mdv1, and Dnm1. Therefore, the present work reveals an uncharacterized molecular mechanism underlying the inhibition of mitochondrial fission.

**Funding:** This work is supported by grants from the National Key Research and Development Program of China (https://service.most.gov.cn/) (2022YFA1303100 to X.Y.), the National Natural Science Foundation of China (https://www.nsfc.gov.cn/) (91754106, 32070707, and 31621002 to C.F. and 31872759 to A.T.), the Center for Advanced Interdisciplinary Science and Biomedicine of IHM (QYPY20220003 to C.F.), and the Fundamental Research Funds from University of Science and Technology of China (WK9110000141 to A.T.). The funders had no role in study design, data collection and analysis, decision to publish, or preparation of the manuscript.

**Competing interests:** The authors have declared that no competing interests exist.

**Abbreviations:** Co-IP, co-immunoprecipitation; EGS, ethylene glycol bis(succinimidyl succinate; EMCCD, electron multiplying charge-coupled device; ER, endoplasmic reticulum; ERAD, ER-associated degradation; GRIP1, glutamate receptor-interacting protein 1; MBP, maltose-binding protein; mitoCPR, mitochondrial compromised protein import response; TA, tail-anchored; WT, wild-type.

## Introduction

Tail-anchored (TA) proteins are involved in many cellular functions, including protein translocation, organelle dynamics, and vesicle trafficking [1]. A single transmembrane domain is present at the C-termini of TA proteins to allow insertion of the proteins into organelle membranes [2]. The endoplasmic reticulum (ER) is one of the main destinations for TA proteins [2]. When the routes for targeting TA proteins to the ER are impaired, the TA proteins could be mistargeted to the mitochondrial outer membrane, risking the impairment of mitochondrial function [3,4]. Within the cell, Msp1 (ATAD1 in humans) functions to clear the mistargeted TA proteins on mitochondria and to safeguard mitochondrial quality [5–7].

Msp1/ATAD1 belongs to the AAA-ATPase family characterized by the presence of an AAA-ATPase domain capable of forming oligomers [5]. A single transmembrane domain is present at the N-terminus to allow insertion of Msp1/ATAD1 into the mitochondrial outer membrane. Over the past several years, the function and structure of Msp1 have been characterized intensively, establishing that Msp1 functions as a hexameric translocase to extract mistargeted TA proteins from the mitochondrial outer membrane [3,4,8–12]. It is also increasingly clear that Msp1 collaborates with the ER-associated degradation (ERAD) system to clear TA proteins that are mistargeted to the mitochondrial outer membrane [13,14]. Most of the above breakthroughs are based on using the peroxisomal protein Pex15 (PEX26 in humans) as a model client protein of Msp1. Currently, in addition to Pex15, a limited number of TA proteins have been identified as Msp1 client proteins: Gos1 (GOS28 in humans) [4], Fmp32 [14,15], Frt1 [11], and Ysy6 [11]. Among these client proteins, only Pex15 appears to contain a hydrophobic segment that can be recognized by Msp1 [11]. In addition to the model, in which Msp1 recognizes substrates through the characteristic hydrophobic segment, 2 other possible models have been proposed. One model proposes that a shielding protein interacts with the client proteins to block the access of Msp1 [16], while the other proposes that Msp1 is capable of extracting only orphan monomeric proteins due to energy limitation [14]. How Msp1 interacts with its client proteins remains to be further investigated.

Msp1/ATAD1 does not interact with only TA proteins because ATAD1 also interacts with the non-TA GluR2-containing protein AMPAR to disassemble AMPAR from GRIP1 (glutamate receptor-interacting protein 1) [17]. Moreover, Msp1 interacts with Cis1, a non-TA cytoplasmic protein, to be involved in the mitochondrial compromised protein import response (mitoCPR) [18], and ATAD1 interacts with non-TA proteins PLAA and UBXN4 to promote the degradation of desmin intermediate filaments [19]. Hence, Msp1/ATAD1 may be a versatile molecule that interacts with diverse client proteins to mediate a wide range of cellular functions.

In addition to governing mitochondrial quality on the mitochondrial outer membrane, Msp1/ATAD1 appears to be involved in regulating mitochondrial morphology. In ATAD1[-/-] mouse embryonic fibroblasts or ATAD1-knockdown HeLa cells, mitochondria are fragmented [4]. The absence of only Msp1 in budding yeast appears to affect mitochondria slightly, but the absence of Msp1 and the GET (guided entry of tail-anchored proteins) system component Get1, Get2, or Get3 alters mitochondrial morphology significantly [3,4]. What is the specific role of Msp1/ATAD1 in regulating mitochondrial morphology under physiological conditions? Does Msp1/ATAD1 interact with mitochondrial fission factors to regulate mitochondrial dynamics? These questions remain to be addressed.

The fission yeast *Schizosaccharomyces pombe* depends on microtubules for positioning mitochondria within the cell, while the budding yeast *Saccharomyces cerevisiae* mainly relies on actin filaments to distribute mitochondria [20]. By contrast, mammalian cells use both microtubules and actin filaments to distribute mitochondria. Therefore, fission yeast is an

excellent model organism complementing budding yeast for studying mitochondrial dynamics. The proteins regulating mitochondrial dynamics have been conserved through evolution. For example, the dynamin-related GTPase Dnm1 (Drp1 in humans) is responsible for mitochondrial fission, and the dynamin-like GTPases Fzo1 (Mfn1 and Mfn2 in humans) and Mgm1 (Opa1 in humans) are responsible for fusion of the mitochondrial outer and inner membranes, respectively [21–23]. Moreover, while multiple receptor proteins (e.g., Mff, MiD49, and MiD51) recruit Drp1 to the mitochondrial outer membrane in mammalian cells, only 1 receptor protein, i.e., Fis1, recruits the Drp1 counterpart Dnm1 to the mitochondrial outer membrane via Caf4/Mdv1 in yeasts [24]. Hence, in yeasts, Fis1, Mdv1, and Dnm1 are involved in forming the mitochondrial divisome for mediating mitochondrial fission [25]. In response to different metabolic stimuli or intracellular signaling, mitochondria undergo remodeling by posttranslationally regulating Drp1/Dnm1 and/or its receptor proteins [26]. Whether Msp1/ATAD1 regulates mitochondrial dynamics by directly modulating the function of Drp1/Dnm1 and/or its receptor proteins has not been tested.

In this study, we demonstrate that the fission yeast counterpart of Msp1/ATAD1, i.e., Yta4, prevents mitochondrial fission by interacting with the mitochondrial divisome components Fis1, Mdv1, and Dnm1. The absence of Yta4 promotes Dnm1-dependent mitochondrial fission, while Yta4 overexpression inhibits mitochondrial fission by impairing the localization of Fis1, Mdv1, and Dnm1 to mitochondria. Moreover, the absence of Yta4 sensitizes mitochondria to the treatment with FCCP, a potent uncoupler of mitochondrial oxidative phosphorylation, enhancing mitochondrial fragmentation. Hence, in this study, we present a Yta4-dependent mechanism underlying the dynamic control of mitochondrial fission.

## Results

### The absence of Yta4 causes mitochondrial fragmentation

Mouse embryonic fibroblasts lacking ATAD1 display fragmented mitochondria [4]. By contrast, the absence of the ATAD1 homolog Msp1 in budding yeast appears to affect mitochondrial morphology only slightly [3,4]. Yta4 is the homolog of Msp1/ATAD1 in fission yeast. Using fission yeast as a model organism, we examined mitochondrial morphology in wild-type (WT) and *yta4*-deleted (*yta4Δ*) cells. First, we directly stained WT and *yta4Δ* cells with MitoTracker Red (Fig 1A) and quantified the number of mitochondria, mitochondrial branches, and mitochondrial junctions (Fig 1B). It was apparent that in *yta4Δ* cells, the mitochondrial number increased, whereas the number of mitochondrial branches and junctions decreased significantly, suggesting that the absence of Yta4 causes mitochondrial fragmentation and reduces the complexity of the mitochondrial network (Fig 1B). For ease of observation by live-cell microscopy, we tagged Sdh2, a mitochondrial matrix protein [27], at its own locus with mCherry in WT and *yta4Δ* cells or *yta4Δ* cells expressing Yta4-13Myc from its own promoter. As shown in Fig 1C and 1D, the absence of Yta4 similarly caused mitochondrial fragmentation in these Sdh2-mCherry-expressing cells, while ectopic expression of Yta4-13Myc in *yta4Δ* cells restored mitochondria to wild-type-like tubular morphology (Fig 1C). The expression of Yta4-13Myc was confirmed by western blotting analysis (Fig 1E). Although a similar phenotype of mitochondrial fragmentation was observed in MitoTracker-stained and Sdh2-mCherry-expressing *yta4Δ* cells, we noticed that tagging Sdh2 appeared to alleviate the phenotype of mitochondrial fragmentation caused by the absence of Yta4. Analysis by live-cell microscopy further showed that both fission and fusion frequencies increased significantly in cells lacking Yta4 and that ectopic expression of Yta4 rescued the phenotype (Fig 1F and 1G). Together, these data indicate that Yta4 may function to inhibit mitochondrial dynamics to maintain tubular mitochondrial structures.

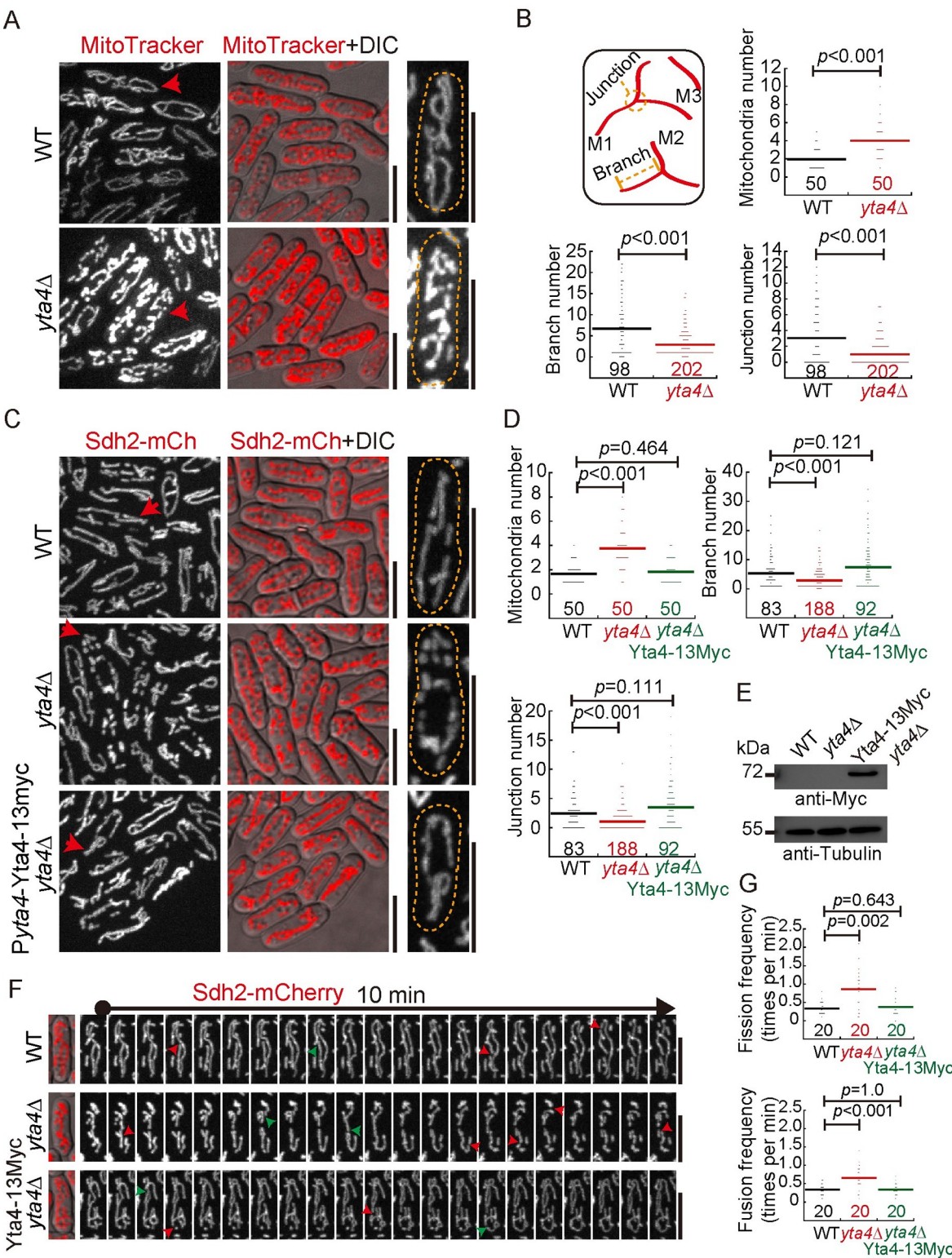

**Fig 1. Mitochondrial morphology and dynamics in WT and *yta4*-deleted cells.** (A) Maximum projection images of WT and *yta4Δ* (*yta4* deletion) cells stained with MitoTracker Red. The magnified images on the right are the cells indicated by red arrows, and dashed lines mark the edge of the cells. DIC, differential interference contrast. Scale bars: 10 μm. (B) Quantification of the mitochondrial number (per cell), branch (per cell), and junction (per cell) of WT and *yta4Δ* cells shown in (A) using the algorithm MiNA in ImageJ. The diagram on the left illustrates the mitochondrial number, branch, and junction (red lines indicate 3 mitochondria: M1, M2, and M3). The

Wilcoxon–Mann–Whitney rank sum test was used to calculate *p* values, the horizontal line indicates the mean, and the number of cells analyzed is shown on the x-axis. Experiments were repeated twice, and raw data are available in S1 Data. (C) Maximum projection images of WT and *yta4Δ* cells expressing Sdh2-mCherry (a mitochondrial matrix protein). The magnified images on the right are the cells indicated by red arrows, and dashed lines mark the edge of the cells. Scale bars: 10 μm. (D) Quantification of the mitochondrial number, branch, and junction of the Sdh2-mCherry-expressing WT and *yta4Δ* cells and *yta4Δ* cells carrying Yta4-13Myc using the algorithm MiNA in ImageJ. The Wilcoxon–Mann–Whitney rank sum test was used to calculate *p* values (vs. WT), the horizontal line indicates the mean, and the number of cells analyzed is shown on the x-axis. Experiments were repeated twice, and raw data are available in S1 Data. (E) Testing the expression of Yta4-13Myc by western blotting. Antibodies against Myc and Tubulin were used. (F) Maximum projection time-lapse images of WT and *yta4Δ* cells expressing Sdh2-mCherry (also see S1–S3 Movies). Red and green arrowheads mark mitochondrial fission and fusion, respectively. Scale bars: 10 μm. (G) Quantification of the mitochondrial fission (top) and fusion (bottom) frequencies of WT and *yta4Δ* cells. Time-lapse images indicated in (F) were used for analysis. The *p* values of fission and fusion frequency were calculated by the Wilcoxon–Mann–Whitney rank sum test (vs. WT) and one-way ANOVA (with post hoc Tukey honest significance difference test), respectively. The horizontal line indicates the mean, and the number of cells analyzed is shown on the x-axis. Experiments were repeated twice, and raw data are available in S1 Data. WT, wild-type.

## The absence of Yta4 enhances FCCP-induced mitochondrial fragmentation

Mitochondrial respiration increases upon treatment of cells with a potent uncoupler of mitochondrial oxidative phosphorylation, i.e., FCCP [28]. Therefore, FCCP treatment mimics a condition of enhanced mitochondrial respiration. We then asked whether the absence of Yta4 sensitizes mitochondria to fragmentation upon FCCP treatment. To address this question, we employed profusion chambers to monitor the morphological changes in mitochondria by live-cell microscopy. As shown in Fig 2A and 2B, after 4 min of treatment with 0.5 μm FCCP, mitochondria further fragmented in *yta4Δ* cells, but not in WT cells. Treatment with a higher concentration of FCCP (e.g., 2 μM) induced rapid mitochondrial fragmentation in both *yta4Δ* and WT cells (Fig 2C). These data suggest that a Yta4-dependent mechanism exists within the cell to prevent mitochondrial fragmentation in response to elevated mitochondrial respiration or mild mitochondrial depolarization.

## The absence of Yta4 causes mitochondrial fragmentation in a Dnm1-dependent manner

We further hypothesized that mitochondrial fragmentation caused by the absence of Yta4 depends on Dnm1 since Dnm1 is the master regulator of mitochondrial fission. To test this hypothesis, we deleted either *dnm1* alone (*dnm1Δ*) or both *dnm1* and *yta4* (*yta4Δdnm1Δ*) in Sdh2-mCherry-expressing cells. Consistently, mitochondria became fragmented in *yta4Δ* cells but not in WT cells after treatment with 0.5 μM FCCP, and mitochondria became fragmented in both *yta4Δ* and WT cells after treatment with 2 μM FCCP (Fig 3A). By contrast, mitochondria formed a similar tubular interconnected structure in both *dnm1Δ* and *yta4Δdnm1Δ* cells with or without FCCP treatment (Fig 3A). We noticed that mitochondria underwent constriction, but not fission, in *yta4Δdnm1Δ* cells after the treatment with 2 μM FCCP (Fig 3A, bottom panel). Since the wrapping of the ER on mitochondria mediates mitochondrial constriction and promotes the Dnm1-dependent mitochondrial fission [29], we speculated that mitochondrial constriction may be due to the wrapping of the ER on mitochondria, which may be regulated by Yta4. Collectively, the results suggest that mitochondrial fragmentation caused by the absence of Yta4 depends on Dnm1

To test whether the mitochondrial fragmentation was due to the altered expression of Dnm1, we examined the expression of Dnm1 by western blotting. Quantification showed that the expression levels of endogenous Dnm1 were comparable in WT and *yta4Δ* cells (Fig 3B). Therefore, mitochondrial fragmentation in *yta4Δ* cells is not due to altered expression of Dnm1.

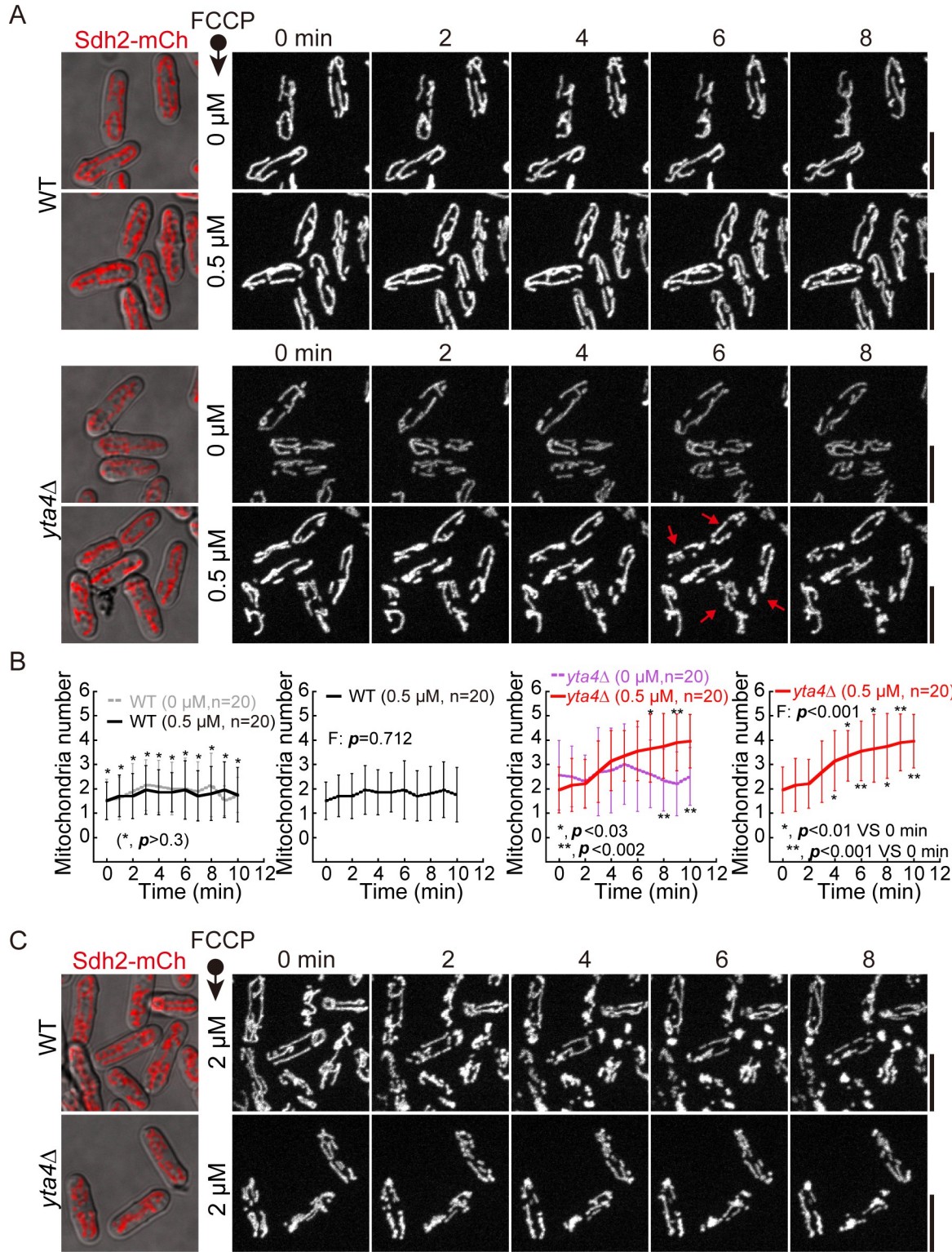

**Fig 2. FCCP-induced mitochondrial fragmentation in WT and *yta4Δ* cells.** (A) Maximum projection time-lapse images of WT and *yta4Δ* cells expressing Sdh2-mCherry (also see S4–S7 Movies). Cells treated with FCCP at the indicated concentrations (0 μM and 0.5 μM) in profusion chambers were imaged immediately after the treatment. Scale bars: 10 μm. (B) Quantification of the mitochondrial number of WT and *yta4Δ* cells that were or were not treated with FCCP. Time-lapse movies as indicated in (A) were used for the quantification, and 20 cells at each time point were analyzed. Data represent mean ± SD. For the comparison of the effect of 2 treatments

(i.e., 0 or 0.5 μM) on WT or *yta4Δ* cells, the Wilcoxon–Mann–Whitney rank sum test was performed to calculate the *p* values of the 2 sets of values at each time point. To test the effect of 0.5 μM FCCP treatment on WT or *yta4Δ* cells over time, Friedman test was performed to calculate the *p* values (indicated as *F: p* in the graphs). In addition, in the right graph (*yta4Δ* cells treated with 0.5 μM FCCP), the Wilcoxon–Mann–Whitney rank sum test was performed to calculate the *p* values of the comparisons between 0-min and other time points. Experiments were repeated twice, and raw data are available in S1 Data. (C) Maximum projection time-lapse images of Sdh2-mCherry-expressing WT and *yta4Δ* cells treated with 2 μM FCCP. Note that mitochondria fragmented rapidly both in WT and *yta4Δ* cells after the cells were treated with 2 μM FCCP. Scale bars: 10 μm. WT, wild-type.

We then examined the localization of Dnm1. Previously, we showed that tagging Dnm1 either N-terminally or C-terminally at its own locus compromises its function [30]. Nonetheless, ectopically expressing Dnm1-GFP (from the *ase1* promoter) at the *leu1* locus in cells containing endogenous Dnm1 allows the visualization of Dnm1 and maintains proper mitochondrial dynamics [30]. In the present study, we employed a similar approach to express Dnm1-GFP in WT and *yta4Δ* cells carrying Sdh2-mCherry. As shown in Fig 3C, the absence of Yta4 did not affect the localization of Dnm1-GFP to mitochondria. In addition, we performed live-cell imaging to assess the fission activity of Dnm1. WT and *yta4Δ* cells were treated with or without 0.5 μM FCCP, and images were acquired every half a minute for 10 min (Fig 3D). Quantification showed that the percentage of Dnm1 foci associated with mitochondrial fission increased significantly in *yta4Δ* cells and increased further upon FCCP treatment (Fig 3E). Collectively, we concluded that the absence of Yta4 does not affect the localization of Dnm1 to mitochondria but promotes mitochondrial fission, likely by enhancing the fission activity of Dnm1.

## Overexpression of Yta4 impairs the localization of Dnm1 and Fis1 to mitochondria

The above findings indicate that Yta4 may function as an inhibitory factor of Dnm1. To further test this idea, we overexpressed WT and 2 specific mutants (an ATP hydrolysis-deficient mutant and a translocation-deficient mutant) of Yta4 in *yta4Δ* cells. The 2 specific mutants of the counterpart of Yta4 in budding yeast (i.e., Msp1) have been reported and characterized [8]. Therefore, we mutated the conserved amino acid residues in Yta4 to generate the 2 mutants (Fig 4A) and referred to the ATP hydrolysis-deficient mutant as Yta4(EQ) (Glu192, mutated to Gln192) and the translocation-deficient mutant as Yta4(AA) (Trp165Phe166, mutated to Ala165Ala166). We then followed the experimental procedure in S1A Fig to test the expression of Yta4 and the effects of Yta4 overexpression on the localization of Dnm1 and Fis1 to mitochondria.

The proper expression of WT Yta4 and the Yta4 mutants was confirmed by western blotting assays (Figs 4B, 4C and S1B). As shown in Figs 4D and S1C, overexpression of WT Yta4 impaired the formation of Dnm1 foci on mitochondria but did not change the tubular mitochondrial morphology. Similarly, overexpression of Yta4(AA) or Yta4(EQ) impaired the formation of Dnm1 foci on mitochondria but unexpectedly caused mitochondria to aggregate (Figs 4D and S1C). Time-course release experiments (S1A Fig) exhibited that the signals of Dnm1-GFP on mitochondria decreased gradually as the levels of Yta4/Yta4(EQ)/Yta4(AA) increased (S1B and S1C Fig). We speculated that Yta4(EQ) and Yta4(AA) may stably associate with their neighboring TA client proteins that reside on the mitochondrial outer membrane, and consequently, the stable association may promote fusion/clustering of neighboring mitochondria and ultimately lead to mitochondrial aggregation. Together, these data indicate that Yta4 inhibits Dnm1 accumulation on mitochondria in an ATPase/translocase-independent manner.

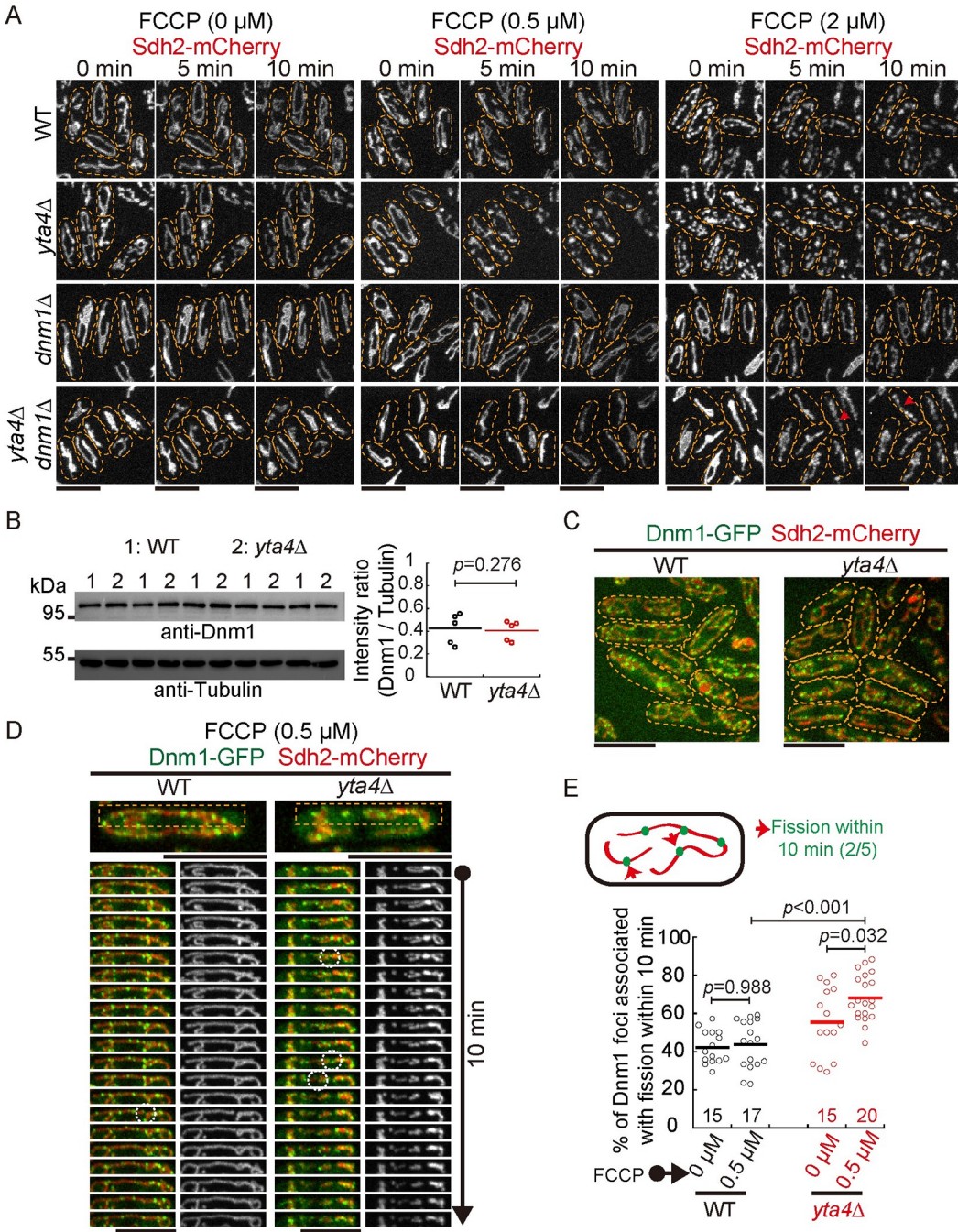

**Fig 3. Dnm1 is required for promoting mitochondrial fragmentation in *yta4Δ* cells.** (A) Maximum projection time-lapse images of Sdh2-mCherry-expressing WT, *yta4Δ*, *dnm1Δ*, and *yta4Δdnm1Δ* cells that were or were not treated with 0.5 or 2 μM FCCP. Dashed lines mark the edge of cells. Mitochondria displaying incomplete constriction are marked by red arrows. Scale bars: 10 μm. (B) Testing Dnm1 expression by western blotting. Protein extracts from 5 WT (indicated by 1) and *yta4Δ* (indicated by 2) samples were analyzed with antibodies against Dnm1 and Tubulin. Quantification of the intensity ratio of Dnm1 to Tubulin is shown on the right. The horizontal line indicates the mean, and statistical analysis was performed by Student's *t* test. Raw data are available in S1 Data. (C) Maximum projection images of WT and *yta4Δ* cells that express Sdh2-mCherry from its own promoter and ectopically express Dnm1-GFP from the *ase1* promoter. Dashed lines mark the edge of cells. Scale bars: 10 μm. (D) Maximum projection time-lapse images of the indicated cells in (C). Note that cells were treated with 0.5 μM FCCP. Montage images of the dashed line-marked region on the WT and *yta4Δ* cells are shown, and dashed circles indicate Dnm1-associated fission sites. Scale bars: 10 μm. (E) Quantification of the percentage of Dnm1-associated mitochondrial fission. WT and *yta4Δ* cells were treated with or without 0.5 μM FCCP. The diagram illustrates that 2 Dnm1-associated mitochondrial fission events (marked by the red arrows) take place within 10 min of

microscopic observation. Statistical analysis was performed by one-way ANOVA with Tukey's honest significance difference test. The horizontal line indicates the mean, and the number of cells analyzed is shown on the x-axis. Experiments were repeated twice, and raw data are available in S1 Data. WT, wild-type.

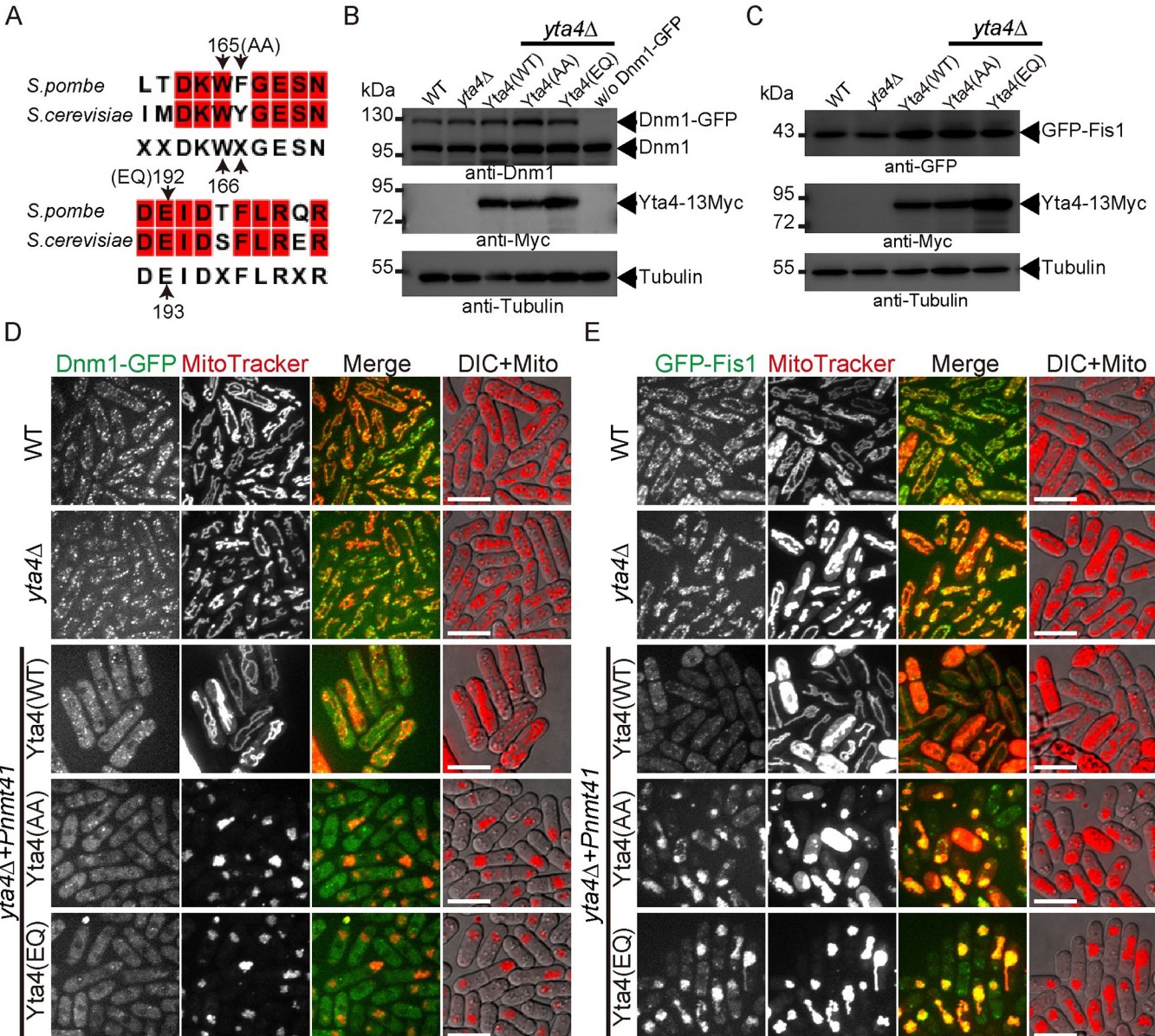

**Fig 4. The effect of Yta4 overexpression on the localization of Dnm1 and Fis1 to mitochondria.** (A) Alignment of the amino acid sequences of Yta4 and its budding yeast homolog Msp1. The conserved residues are highlighted in red, and the residues mutated (WF and E, indicated in the brackets, were mutated into AA and Q, respectively) in Yta4 for functional analysis are indicated by arrows. (B) Testing the expression of Dnm1-GFP (ectopically expressed from the *ase1* promoter), endogenous Dnm1, and different Yta4-13Myc variants (expressed ectopically) in the indicated cells used in (D) by western blotting with antibodies against Dnm1, Myc, and Tubulin. (C) Testing the expression of GFP-Fis1 (Fis1 was tagged at its own locus at the 5′ end and expressed from the *ase1* promoter) and different Yta4-13Myc variants (expressed ectopically) in the indicated cells used in (E) by western blotting with antibodies against GFP, Myc, and Tubulin. (D) Maximum projection images of Dnm1-GFP-expressing WT and *yta4Δ* cells and Dnm1-GFP-expressing *yta4Δ* cells that carry the indicated variants of Yta4-13Myc (from the *nmt41* promoter). Cells were cultured in EMM5S medium lacking thiamine, and mitochondria were stained with MitoTracker Red. Scale bars: 10 µm. (E) Maximum projection images of GFP-Fis1-expressing WT and *yta4Δ* cells and GFP-Fis1-expressing *yta4Δ* cells that carry the indicated variants of Yta4-13Myc (from the *nmt41* promoter). Cells were cultured in EMM5S medium lacking thiamine, and mitochondria were stained with MitoTracker Red. Scale bars: 10 µm. DIC, differential interference contrast; WT, wild-type.

Interestingly, overexpression of Yta4, but not Yta4(AA) and Yta4(EQ), delocalized Fis1 (Figs 4E and S1D), the mitochondrial TA protein responsible for recruiting Dnm1 to mitochondria. In addition, time-course release experiments exhibited that the delocalization of Fis1 from mitochondria was quite efficient at only 6 h after the overexpression of Yta4 (S1D Fig). Examination of the localization of GFP-Fis1 in cells expressing Ost4-tdTomato, a proxy protein marking the ER, showed that the delocalized GFP-Fis1 appeared to be present in the cytoplasm of Yta4-overexpressing cells, while GFP-Fis1 in Yta4(AA)-overexpressing or Yta4 (EQ)-overexpressing cells was still present on mitochondria, which were clearly separated from the ER (S1D and S2 Figs). This result is consistent with the previous finding that the ATPase activity and the key residues within the loop 1 of Msp1/ATAD1 are required for extracting TA proteins [8]. Collectively, these results suggest that Yta4 functions to inhibit mitochondrial fission through regulating Dnm1 and Fis1.

We further asked how Yta4 and its mutants (i.e., Yta4(AA) and Yta4(EQ)) that were expressed at the endogenous level affect mitochondria. To address this question, we expressed Yta4(WT) and its mutants from the *yta4* promoter in *yta4Δ* cells (S3A–S3C Fig). The expression of Yta4(WT) and its mutants was confirmed by a western blotting analysis (S3B Fig), and we noticed that the expression levels of Yta4(WT) and Yta4(EQ) were comparable but were higher than the expression level of Yta4(AA). Microscopic observation showed that the localization of both GFP-Fis1 and Dnm1-GFP to mitochondria did not appear to be affected by the expression of Yta4(WT), Yta4(AA), or Yta4(EQ) (S3A and S3C Fig). Consistent with the data shown in Fig 1C, the expression of Yta4(WT) restored mitochondria to tubular structures in *yta4Δ* cells (S3A and S3C Fig), and similarly, the expression of Yta4(AA) and Yta4(EQ) restored mitochondria to tubular structures in *yta4Δ* cells (S3A and S3C Fig). Consistently, determination of the percentage of Dnm1 foci that were associated with mitochondrial fission confirmed that the absence of Yta4 enhanced Dnm1-associated mitochondrial fission, while the expression of Yta4(WT), Yta4(AA), or Yta4(EQ) from the *yta4* promoter restored Dnm1-associated mitochondrial fission to the WT level in *yta4Δ* cells (S3D Fig). Notably, Yta4, overexpressed from the *nmt41* promoter, impaired the mitochondrial localization of Dnm1 in an ATPase/translocase-independent manner (Figs 4D and S1C). In addition, the biochemical data (see the results below) showed that the monomeric form of Yta4 (S4B and S4D Fig), which lacks the transmembrane domain and does not have an ATPase activity (S4E Fig), was sufficient to inhibit Dnm1 function. These results indicate that Yta4, at the endogenous level, may prevent excessive mitochondrial fission by mainly inhibiting Dnm1 in an ATPase/translocase-independent manner.

In contrast to the significant effect of Yta4 overexpression on Fis1, the absence of Yta4 did not appear to significantly alter the mitochondrial localization of Fis1 (Fig 4C and 4E). Consistently, quantification showed that the average intensity of GFP-Fis1 signals on mitochondria was not significantly altered by the absence of Yta4 (S3E Fig). These results indicate that under physiological conditions, Yta4 may play a minor role in extracting Fis1 from the mitochondrial outer membrane. However, it is also possible that under physiological conditions, Yta4-mediated extraction of Fis1 takes place less frequently at only a few subregions on mitochondria, which may not be discernible (see Discussion for details).

## Overexpression of Yta4 delocalizes Fis1 and Dnm1 from mitochondria

To further test the effect of Yta4 overexpression on Fis1 and Dnm1, we employed an approach of cell fractionation. Specifically, 2 types of cells were created: one overexpressing Yta4 from the *nmt41* promoter while the other overexpressing the control maltose-binding protein (MBP) from the *nmt41* promoter (Fig 5A and 5C). Mitochondrial and cytosolic fractions of

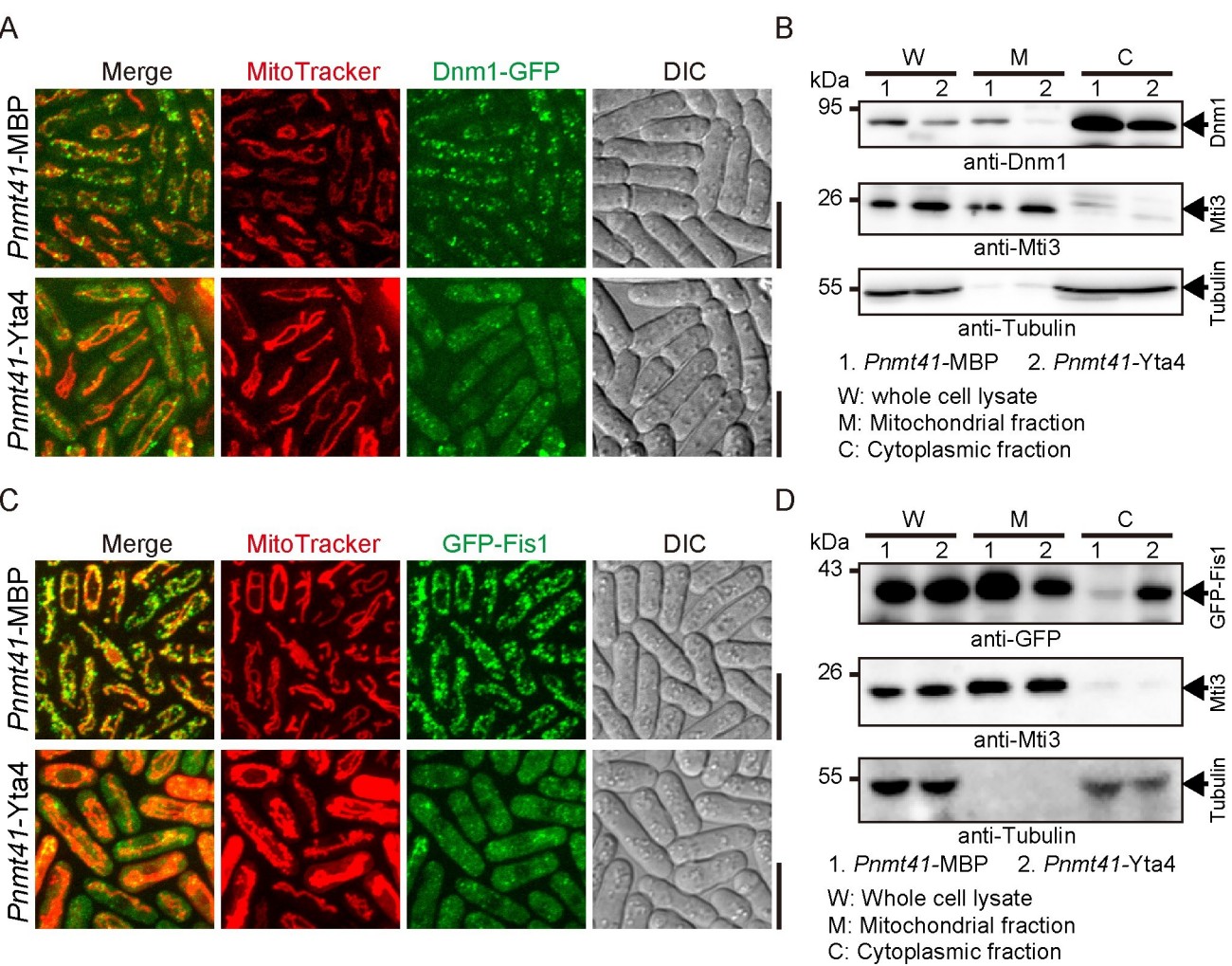

**Fig 5. Determining the distribution of Dnm1 and Fis1 by cell fractionation.** (A) Maximum projection images of Dnm1-GFP-carrying cells overexpressing Yta4 or MBP from the *nmt41* promoter. Cells were cultured in EMM5S medium lacking thiamine, and mitochondria were stained with MitoTracker Red. Scale bars: 10 μm. (B) Mitochondrial and cytosolic fractions were prepared using the cells indicated in (A). *W*, *M*, and *C* indicate whole cell lysate, the mitochondrial fraction, and the cytosolic fraction, respectively. Western blotting assays were performed with antibodies against Dnm1, Mti3 (a mitochondrial protein residing in the matrix), and Tubulin (representing the cytosolic fraction). Note almost no Dnm1 was detected in the mitochondrial fraction of Yta4-overexpressing cells. (C) Maximum projection images of GFP-Fis1-carrying cells overexpressing Yta4 or MBP from the *nmt41* promoter. Cells were cultured in EMM5S medium lacking thiamine, and mitochondria were stained with MitoTracker Red. Scale bars: 10 μm. (D) Mitochondrial and cytosolic fractions were prepared using the cells indicated in (C). *W*, *M*, and *C* indicate whole cell lysate, the mitochondrial fraction, and the cytosolic fraction, respectively. Western blotting assays were performed with antibodies against GFP, Mti3, and Tubulin. Note the amount of GFP-Fis1 in the mitochondrial fraction of Yta4-overexpressing cells was significantly decreased and the amount of GFP-Fis1 in the cytosolic fraction of Yta4-overexpressing cells significantly increased. DIC, differential interference contrast; MBP, maltose-binding protein.

the 2 types of cells were prepared and were analyzed by western blotting assays with antibodies against Tubulin (indicating the cytosolic fraction), Mti3 (indicating the mitochondrial fraction), Dnm1, and GFP (indicating GFP-Fis1). As shown in Fig 5B, the amount of Dnm1 in the mitochondrial fraction decreased when Yta4 was overexpressed (note that the cells did not express Dnm1-GFP). This result was consistent with the microscopic observation, in which Dnm1-GFP signals on mitochondria decreased in Yta4-overexpressing cells (Fig 5A). Similarly, the amount of GFP-Fis1 in the mitochondrial fraction decreased when Yta4 was overexpressed (Fig 5D). Moreover, in the cytosolic fraction of MBP-overexpressing cells, almost no GFP-Fis1 was detected, whereas in the cytosolic fraction of Yta4-overexpressing cells, a

significant amount of GFP-Fis1 was detected (Fig 5D). These biochemical data were consistent with the microscopic observation showing that GFP-Fis1 was present in the cytoplasm of Yta4-overexpressing cells, but not in the cytoplasm of MBP-overexpressing cells (Fig 5C). Hence, it is likely that upon overexpression, Yta4 extracts the TA protein Fis1 from mitochondria and impairs the localization of Dnm1 to mitochondria.

## Yta4 interacts with Dnm1 and Fis1 in vivo

Next, we tested the potential interaction among Yta4, Fis1, and Dnm1 by assessing their colocalization. First, WT cells expressing only Yta4-tdTomato were stained with MitoTracker Green and imaged by live-cell spinning-disk microscopy. As shown in Fig 6A, Yta4-tdTomato decorated the entire mitochondrial network, but Yta4-tdTomato signals were not homogenous on mitochondria, and some subregions showed much stronger staining than other parts on the same mitochondria. Moreover, the colocalization between GFP-Fis1 and Yta4-tdTomato was apparent (Fig 6A), and Yta4-tdTomato appeared to be enriched near the mitochondrial sites where Dnm1-GFP was localized (Fig 6A). The expression levels of Yta4-tdTomato were comparable in the strains shown in Fig 6A and 6B. To test the protein–protein interaction, we employed co-immunoprecipitation (Co-IP) assays. As shown in Fig 6C, GFP-Fis1 precipitated both Yta4-13Myc and endogenous Dnm1. When endogenous Dnm1 was used as a bait, it also precipitated Yta4-13Myc and GFP-Fis1 (Fig 6C). These results suggest that Yta4 interacts with Dnm1 and Fis1 within the cell.

## Overexpression of Yta4 impairs the localization of Mdv1 to mitochondria

Fis1 recruits Dnm1 to the mitochondrial outer membrane via Caf4/Mdv1 [24]. Therefore, Mdv1 bridges Fis1 and Dnm1 [31–35], and the 3 proteins are involved in forming the mitochondrial divisome to promote mitochondrial fission [25]. We further tested whether Yta4 interacts with Mdv1 and regulates the localization of Mdv1 to mitochondria. A similar approach of Yta4 overexpression, as shown in Fig 4, was employed. Live-cell microscopic observation revealed that similar to the effect on Fis1 (Fig 4E), overexpression of Yta4(WT), but not Yta4(AA) and Yta4(EQ), delocalized Mdv1 from mitochondria (Fig 7A and 7B). Note that Mdv1 depends on Fis1 for localizing to mitochondria [34]. Therefore, the similar effects of overexpressed Yta4(WT)/Yta4(EQ)/Yta4(AA) on Fis1 and Mdv1 indicate the nature of tight binding between Fis1 and Mdv1 in vivo. Moreover, GFP-Mdv1 and Yta4-tdTomato colocalized on mitochondria (Fig 7C). In addition, Co-IP assays revealed that Mdv1 interacts with Dnm1 and Yta4 within the cell (Fig 7D). Taken together, these results suggest that Yta4 functions to inhibit mitochondrial fission through regulating Mdv1.

## Yta4 physically interacts with Fis1, Mdv1, and Dnm1 in vitro

We employed GST pull-down assays to test the direct interaction between Yta4 and Dnm1, Mdv1, and Fis1 using recombinant proteins purified from *E. coli*. The domain structures of Fis1, Mdv1, and Dnm1 are illustrated in Fig 8A. In *S. cerevisiae*, the N-terminal arm and the tandem tetratricopepetide repeat-like motifs of Fis1 interact with the N-terminus of Mdv1, which contains the helix–loop–helix motif [31,34,35], while the WD-40 repeats at the C-terminus of Mdv1, which form a seven-bladed propeller, are necessary and sufficient for interacting with Dnm1 [35,36]. Therefore, to preform GST pull-down assays, we separately purified the N-terminus and the C-terminal WD-40 domain of Mdv1 (Fig 8A). As shown in Fig 8B and 8C, GST-Yta4(33–355) (referred to as Yta4(ΔTM), lacking the N-terminal transmembrane domain), but not GST and glutathione resins, precipitated Dnm1 and His-Fis1(1–128) (referred to as Fis1(ΔTM), lacking the C-terminal transmembrane domain). Similarly,

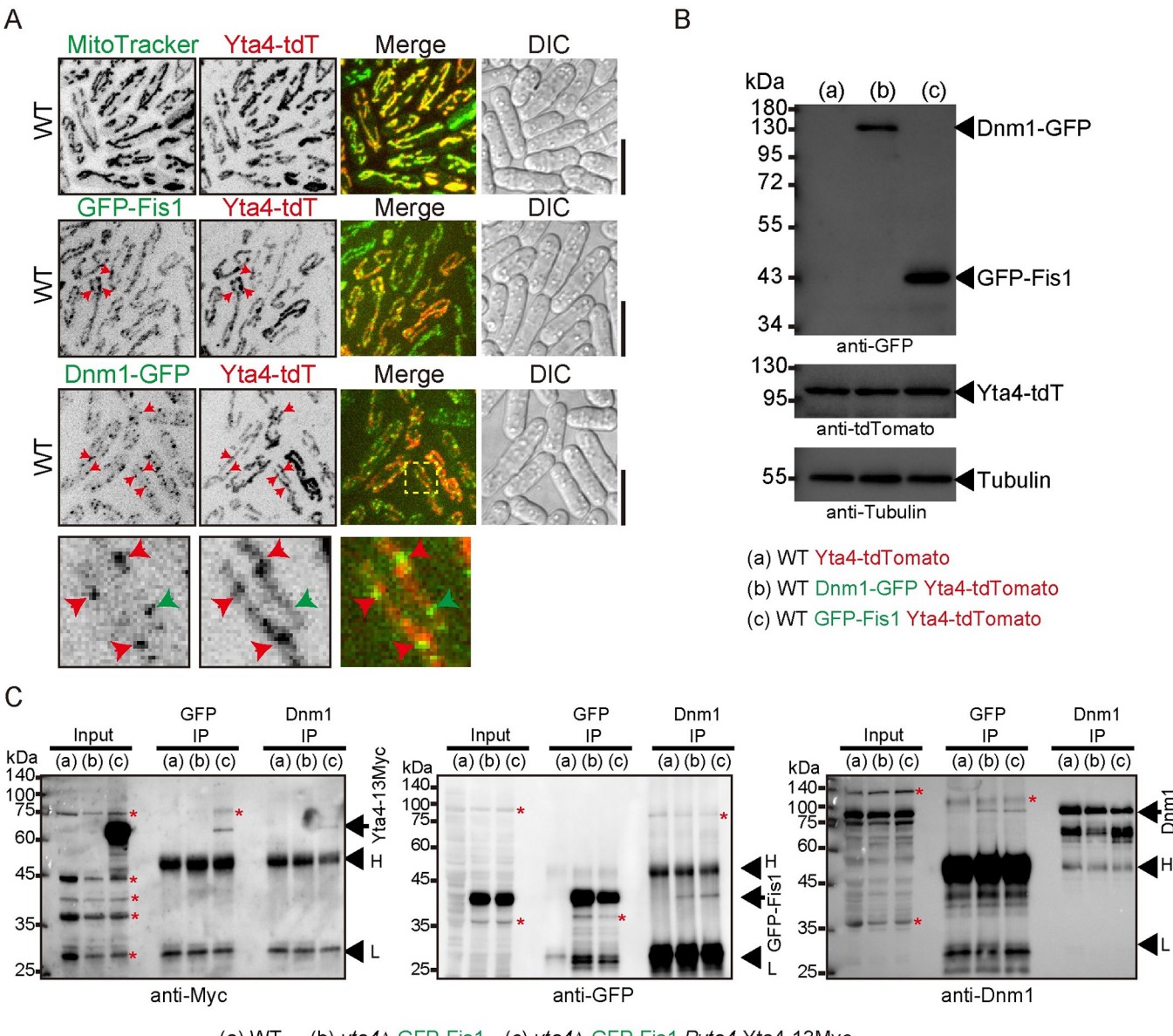

**Fig 6. The interaction and colocalization of Yta4 with Dnm1 and Fis1 in vivo.** (A) Maximum projection images of the indicated cells. Yta4 was tagged with tdTomato, while Fis1 and Dnm1 were tagged with GFP. In the top panel, mitochondria were stained with MitoTracker Green. Red arrows indicate the apposed localization of Yta4-tdTomato with either GFP-Fis1 or Dnm1-GFP. Magnified images at the bottom panel are the region indicated by the yellow dashed square. Note that some Dnm1-GFP foci (indicated by red arrows) are adjacent to and/or colocalized with Yta4-tdTomato foci, while some Dnm1-GFP foci (indicated by green arrows) localized to a mitochondrial region displaying weak Yta4-tdTomato signals. Scale bars: 10 μm. (B) Testing the expression of tag-fused proteins in the indicated cells in (a), (b), and (c) by western blotting with antibodies against GFP, tdTomato, and Tubulin. (C) Co-IP assays for testing the interaction between Dnm1, GFP-Fis1, and Yta4-13Myc. Antibodies against GFP and Dnm1 were used to precipitate GFP-Fis1 and Dnm1, respectively, and the coprecipitates were analyzed by western blotting with antibodies against Myc, GFP, and Dnm1. *H* and *L* indicate heavy and light chains of antibodies, respectively, while arrows indicate Yta4-13Myc, GFP-Fis1, or Dnm1. Red asterisks indicate nonspecific bands. Co-IP, co-immunoprecipitation; DIC, differential interference contrast; WT, wild-type.

GST-Mdv1(319–651) (referred to as Mdv1(WD), i.e., the WD-40 repeats of Mdv1), but not GST and glutathione resins, precipitated Dnm1 and His-Yta4(ΔTM) (Fig 8D and 8E). In addition, the N-terminus of Mdv1 (referred to as Mdv1(NT), i.e., His-Mdv1(1–249)) was co-purified with Fis1(ΔTM)-GST (Fig 8F), suggesting an association of Mdv1(NT) with Fis1.

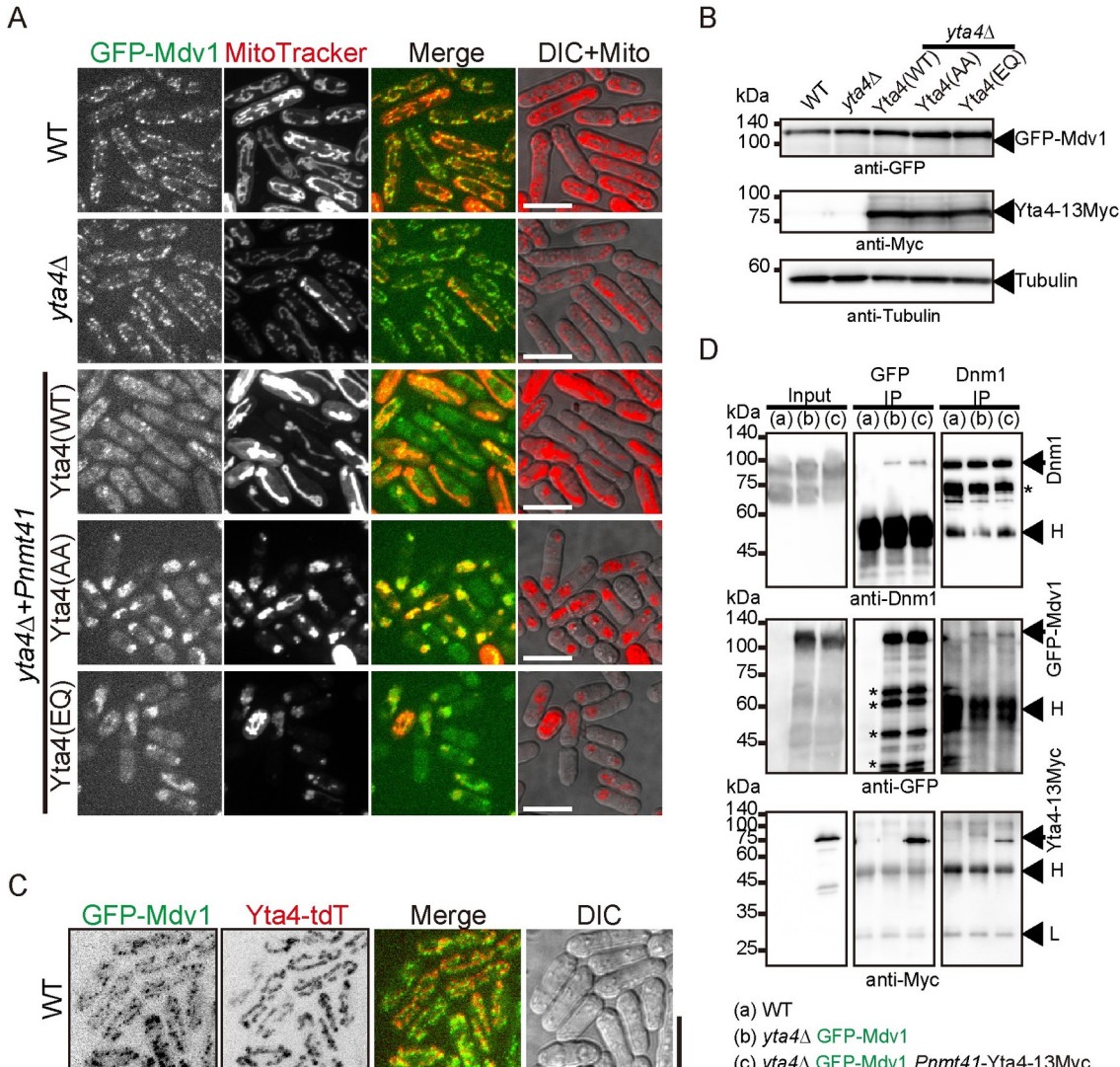

**Fig 7. The effect of Yta4 overexpression on the localization of Mdv1 to mitochondria.** (A) Maximum projection images of GFP-Mdv1-expressing WT and *yta4Δ* cells and GFP-Mdv1-expressing *yta4Δ* cells that carry the indicated variants of Yta4-13Myc (from the *nmt41* promoter). Cells were cultured in EMM5S medium lacking thiamine, and mitochondria were stained with MitoTracker Red. Scale bars: 10 μm. (B) Testing the expression of GFP-Mdv1 and different Yta4-13Myc variants (expressed ectopically) in the indicated cells used in (A) by western blotting with antibodies against GFP, Myc, and Tubulin. (C) Maximum projection images of the indicated cells. Yta4 was tagged with tdTomato while Mdv1 was tagged with GFP. Scale bar: 10 μm. (D) Co-IP assays for testing the interaction between Dnm1, GFP-Mdv1, and Yta4-13Myc. Antibodies against GFP and Dnm1 were used to precipitate GFP-Mdv1 and Dnm1, respectively, and the coprecipitates were analyzed by western blotting with antibodies against Myc, GFP, and Dnm1. *H* and *L* indicate heavy and light chains of antibodies, respectively, while arrows indicate Yta4-13Myc, GFP-Mdv1, or Dnm1. Asterisks indicate degraded forms of GFP-Mdv1. Co-IP, co-immunoprecipitation; DIC, differential interference contrast; WT, wild-type.

Collectively, the above results suggest that Yta4 physically interacts with Dnm1, Mdv1, and Fis1. Given that Mdv1(WD) interacts with both Dnm1 and Yta4, Dnm1 and Yta4 may interact with Mdv1 in a competitive manner (see below).

Msp1, the Yta4 counterpart in budding yeast, forms a hexamer to dislocate its client TA proteins within the cell [35]. The recombinant protein Yta4(ΔTM), used in the above GST pull-down assays, lacks the transmembrane and is mainly monomeric (S4A–S4D Fig). In

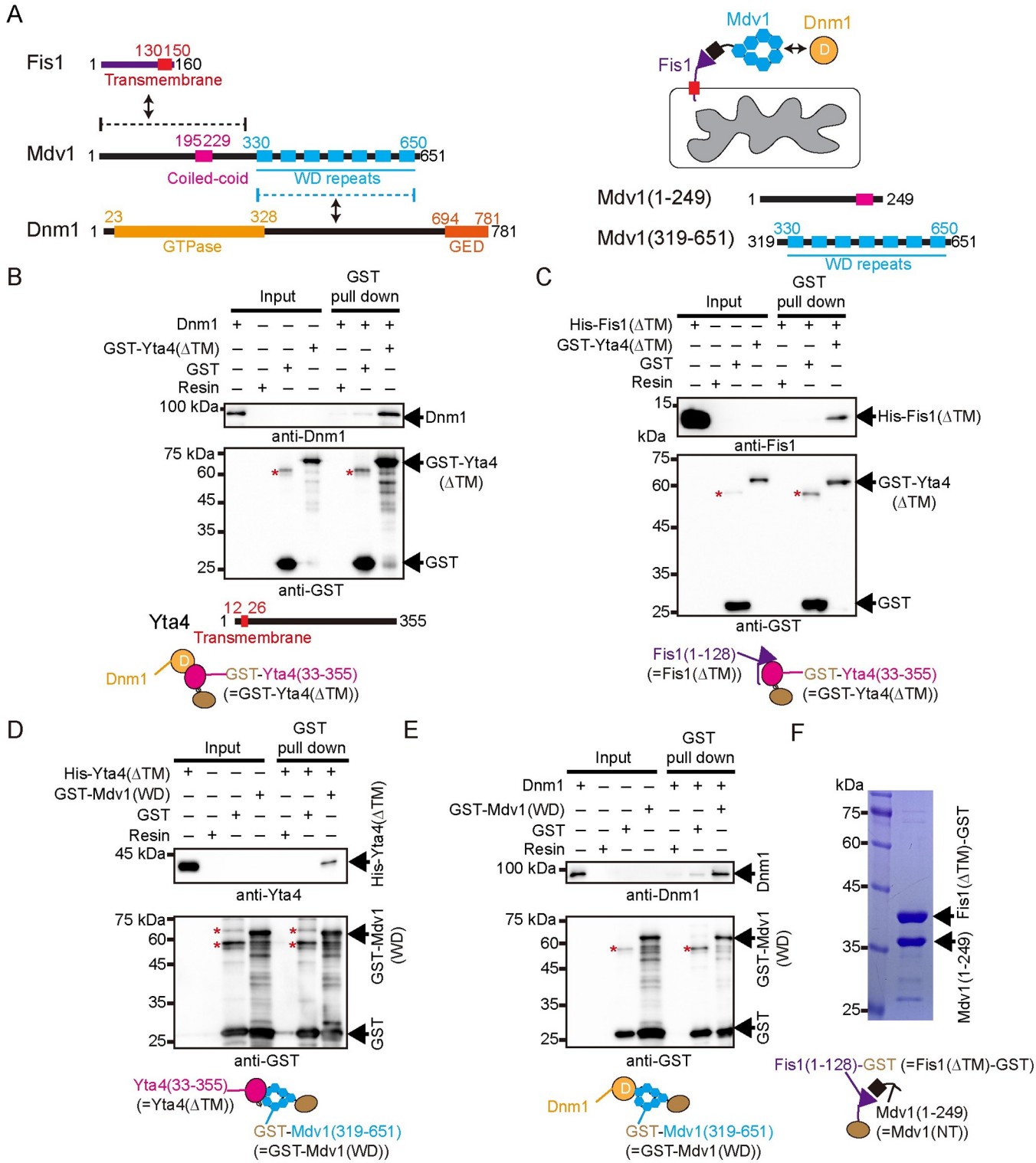

**Fig 8. Testing the physical interactions between Yta4, Dnm1, Fis1, and Mdv1 in vitro.** (A) Diagrams illustrating the domain structures of Fis1, Mdv1, and Dnm1. The diagrams were drawn based on the protein information provided by UniProtKB (https://www.uniprot.org/). Fis1 contains a transmembrane domain at the C-terminus and is inserted into the outer mitochondrial membrane where Fis1 forms a complex (i.e., the mitochondrial divisome) with Mdv1 and Dnm1 to promote mitochondrial fission. WD repeats are present at the C-terminus of Mdv1 while a GTPase domain is present at the N-terminus of Dnm1. The N-termini of Fis1 and Mdv1 interact with one another [31,34], while WD repeats of Mdv1 interacts with Dnm1 [32,35]. (B) GST pull-down assays were performed to test the interaction between Dnm1 and GST-Yta4(33–355) (referred to as GST-Yta4(ΔTM)) or GST (indicated by arrows). Asterisks

indicate nonspecific bands, and the domain structure of Yta4 is shown below the blot. *Resin* indicates a negative control, in which empty resins were used. (C) GST pull-down assays were performed to test the interaction between His-tagged Fis1(1–128) (referred to as His-Fis1(ΔTM)) and GST-Yta4(33–355) or GST (marked by arrows). Asterisks indicate nonspecific bands. *Resin* indicates a negative control, in which empty resins were used. (D) GST pull-down assays were performed to test the interaction between His-Yta4(ΔTM) and GST-Mdv1(319–651) (referred to as GST-Mdv1(WD)) or GST (marked by arrows). Asterisks indicate nonspecific bands. *Resin* indicates a negative control, in which empty resins were used. (E) GST pull-down assays were performed to test the interaction between Dnm1 and GST-Mdv1(WD) or GST (marked by arrows). Asterisks indicate nonspecific bands. *Resin* indicates a negative control, in which empty resins were used. (F) Co-expression and copurification of Fis1(ΔTM)-GST and His-Mdv1(1–249) (referred to as Mdv1(NT)). Protein purification was performed with glutathione resins.

addition, Yta4(ΔTM) did not have an ATPase activity (S4E Fig). Together with the GST pull-down results shown above, these results suggest that the ATPase activity and hexamerization of Yta4 are not necessary for the interactions of Yta4 with Fis1, Mdv1, and Dnm1, at least in vitro.

To further reveal the differential roles of Yta4 monomers and hexamers in regulating client proteins, a hexameric form of Yta4 was produced. A functional hexametic form of Msp1 lacking the transmembrane domain was produced previously by fusing Msp1(ΔTM) to a hexameric scaffold protein Hcp1 [10]. Similarly, we produced a hexameric form of Yta4 by fusing Yta4(ΔTM) to the hexameric scaffold protein Hcp1. Size-exclusion chromatography and ethylene glycol bis(succinimidyl succinate) (EGS) crosslinking experiments revealed that similar to Hcp1-Msp1(ΔTM), Hcp1-Yta4(ΔTM) was a hexamer (S4A–S4D Fig). In addition, ATPase kinetics assays revealed that Hcp1-Yta4(ΔTM), but not the monomeric Yta4(ΔTM), exhibited a significant ATPase activity ($K_m$ = 155.11 μM, $V_{max}$ = 21.38 μM/min) (S4E Fig). We then tested the binding of Yta4(ΔTM) and Hcp1-Yta4(ΔTM) to Dnm1 and Fis1(ΔTM) by GST pull-down assays. As shown in S5A Fig, GST-Hcp1-Yta4(ΔTM) precipitated only slightly more Dnm1 than GST-Yta4(ΔTM), suggesting hexamerization does not appear to have a significant effect on the interaction between Yta4 and Dnm1. Interestingly, in the absence of ATP, GST-Hcp1-Yta4(ΔTM) precipitated much more Fis1(ΔTM) than GST-Yta4(ΔTM), and the presence of ATP significantly decreased the affinity of both GST-Hcp1-Yta4(ΔTM) and GST-Yta4(ΔTM) for Fis1(ΔTM) (S5B Fig). These data suggest that both heximarization and ATP regulate the interaction of Yta4 with Fis1. Given that ATP hydrolysis promotes the motor activity of Msp1/Yta4 [5], we speculated that the dynamic nature of the Yta4 motor domain is the main cause of the weakened interaction between Yta4 and Fis1 in the presence of ATP and in the absence of membranes in vitro. Hence, despite interaction with both Dnm1 and Fis1, Yta4 interacts with them by different mechanisms.

## Yta4 inhibits the interaction of Mdv1 with Dnm1, but not Fis1

To test how Yta4 is involved in regulating the mitochondrial divisome composed of Fis1, Mdv1, and Dnm1, we performed competitive binding assays. Since the interaction between Yta4 and Dnm1 does not depend on the hexamerization of Yta4 and ATP (Figs 8B and S5A), we used the monomeric form of Yta4, i.e., His-Yta4(ΔTM), in the competitive binding assays. As shown in Fig 9A and 9B, the presence of His-Yta4(ΔTM) significantly reduced the affinity of GST-Mdv1(WD) for Dnm1 (to approximately 40% of the original level). This result suggests that Yta4 and Dnm1 interact with Mdv1 in a competitive manner.

Considering that Yta4 also inhibits Dnm1 oligomerization (see the results below), we tested whether the inhibitory effect of Yta4 on the interaction between Dnm1 and Mdv1 (Fig 9A and 9B) was due to the inhibition of Dnm1 oligomerization. We used an assembly-defective mutant of Dnm1 (i.e., Dnm1(G385D)), which was reported previously [37,38]. Sequence alignment exhibited that the Glycine residue at 385 in the *S. cerevisiae* Dnm1 is conserved in the *Schizosaccharomyces pombe* Dnm1 (i.e., Glycine at 380) (S6A Fig). Therefore, to generate

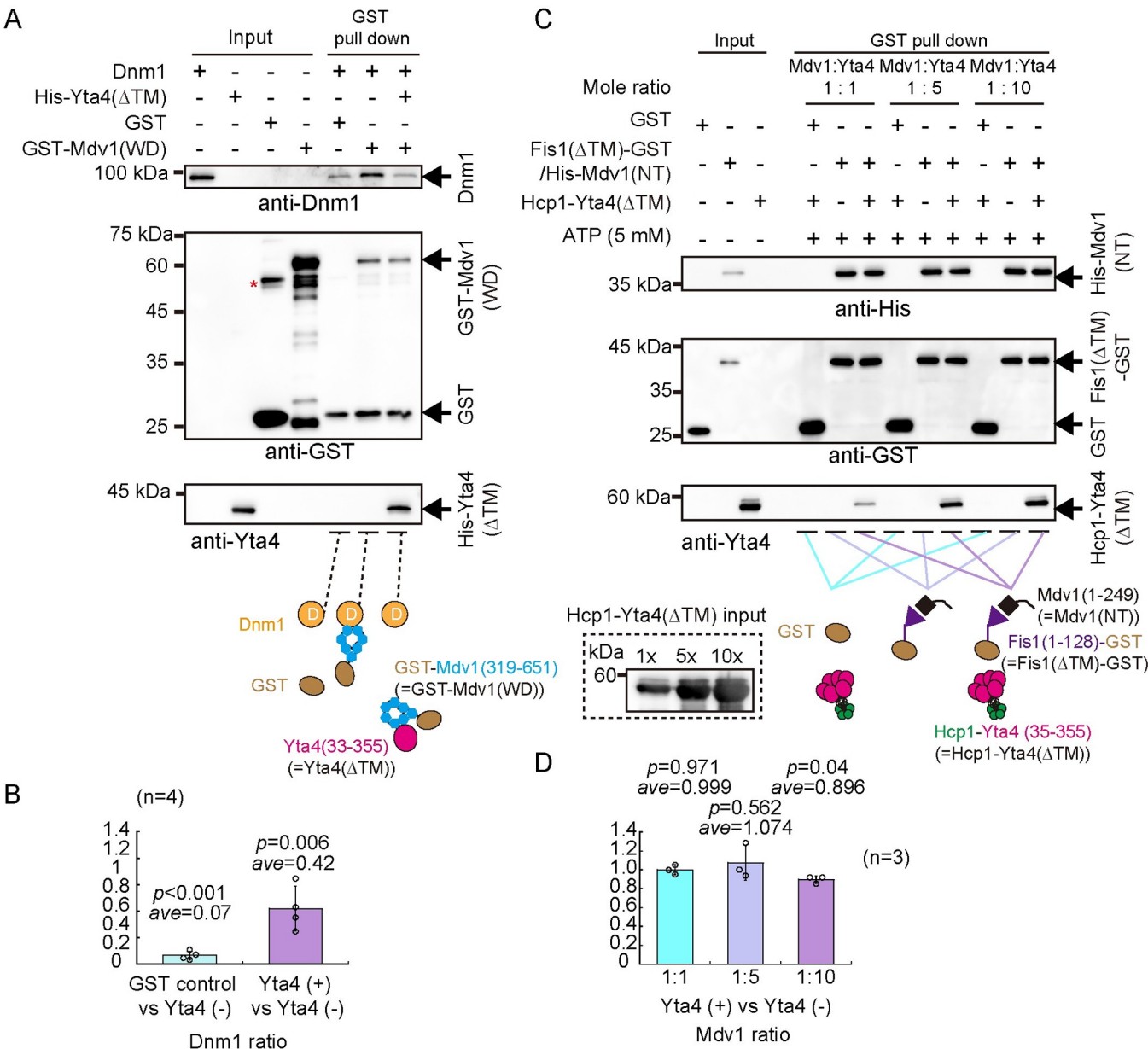

**Fig 9. Testing the competitive effect of Yta4 on the interactions of Mdv1 with Dnm1 and Fis1.** (A) GST pull-down assays were performed to test the interaction between Dnm1 and GST-Mdv1(WD) in the presence or absence of His-Yta4(ΔTM). The asterisk indicates a nonspecific band, and GST was used as a negative control. Note that the presence of His-Yta4(ΔTM) significantly reduced the precipitate of Dnm1 by GST-Mdv1(WD). (B) Quantification of Dnm1 pulled down by GST-fused proteins. The band intensity of Dnm1 was first normalized to the band intensity of corresponding GST proteins, and the ratio of the indicated (x-axis) normalized values of Dnm1 intensity was then calculated. GST control, Yta4(-), Yta4(+) indicate the samples, in which GST, GST-Mdv1 (WD) plus Dnm1, and GST-Mdv1(WD) plus Dnm1 and His-Yta4(ΔTM) were used, respectively. Four independent experiments were performed. The top of the column represents the mean (indicated by *ave*), and single group Student's *t* test was used to calculate the *p* values. Raw data are available in S1 Data. (C) GST pull-down assays were performed to test the interaction between His-Mdv1(NT) and Fis1(ΔTM)-GST in the presence or absence of Hcp1-Yta4(ΔTM). Different concentration ratios (i.e., 1:1, 1:5, and 1:10) of Mdv1 to Yta4 were tested (see the input of Hcp1-Yta4(ΔTM) at the bottom left of the graph). GST was used as a negative control, and 5 mM ATP was added. Note that the presence of Hcp1-Yta4(ΔTM) did not significantly affect the interaction between His-Mdv1(NT) and Fis1(ΔTM)-GST. (D) Quantification of His-Mdv1(NT). The band intensity of His-Mdv1(NT) was first normalized to the band intensity of corresponding GST proteins, and the ratio of the indicated (x-axis) normalized values of His-Mdv1(NT) intensity was then calculated. Yta4(-) and Yta4(+) indicate the samples, in which Fis1(ΔTM)-GST plus His-Mdv1(NT), and Fis1(ΔTM)-GST plus His-Mdv1(NT) and Hcp1-Yta4(ΔTM) were used, respectively. Three independent experiments were performed. The top of the column represents the mean (indicated by *ave*), and single group Student's *t* test was used to calculate the *p* values. Raw data are available in S1 Data.

the *Schizosaccharomyces pombe* version of a Dnm1 assembly-defective mutant, we mutated the Glycine residue at 380 in Dnm1 to aspartic acid (referred to as Dnm1(G380D)). Consistent with the previous finding [37], size-exclusion chromatography confirmed that Dnm1(G380D) was defective in assembly (S6A Fig). Using Dnm1(G380D) and its WT version (referred to as Dnm1(WT)), we performed similar competitive binding assays, as shown in Fig 9A. As showed in S6B and S6C Fig, Dnm1(WT) and Dnm1(G380D) interacted with GST-Mdv1(WD) in a comparable manner (lanes 2 and 5), and His-Yta4(ΔTM) inhibited the interaction of GST-Mdv1(WD) with Dnm1(WT) and Dnm1(G380D) in a comparable manner (lanes 3 and 6). Thus, we concluded that the inhibitory effect of Yta4 on the interaction between Dnm1 and Mdv1 is not due to the inhibition of Dnm1 oligomerization.

Yta4 dislocated Fis1 and Mdv1 from mitochondria in an ATPase/translocase-dependent manner (Figs 4E, 7A and S1D), and Fis1 appeared to have a high affinity for Mdv1 (Fig 8F). Therefore, to test the effect of Yta4 on the interaction between Fis1 and Mdv1, we used the engineered Yta4 hexamer, i.e., Hcp1-Yta4(ΔTM). Interestingly, at the mole ratios of Hcp1-Yta4(ΔTM) to Mdv1(NT) 1:1 or 5:1 and in the presence of 5 mM ATP, Hcp1-Yta4(ΔTM) did not significantly affect the association between Fis1(ΔTM)-GST and His-Mdv1(NT) (Fig 9C and 9D). A further increase in the mole ratio of Hcp1-Yta4(ΔTM) to Mdv1(NT) (10 folds) only slightly weakened the interaction between Fis1(ΔTM)-GST and His-Mdv1(NT) (to approximately 90% of the original level) (Fig 9C and 9D). Similar results were obtained when the monomeric form of Yta4, i.e., His-Yta4(ΔTM), was used in the competitive binding assays (S7A and S7B Fig). These results are consistent with the data showing similar effects of overexpressed Yta4(WT)/Yta4(AA)/Yta4(EQ) on the localization of Fis1 and Mdv1 to mitochondria (Figs 4E and 7A). Hence, we concluded that Yta4 plays a minor role in regulating the interaction between Fis1 and Mdv1.

## Yta4 inhibits the GTPase activity of Dnm1

To understand how Yta4 affects Dnm1 function, we tested the effect of Yta4 on the GTPase activity of Dnm1. First, we purified His-Yta4(ΔTM), which was used in the GST pull-down assays above, and Dnm1 from *Escherichia coli* by affinity and size-exclusion chromatography. After the purity and concentration of the proteins were determined by SDS-PAGE analysis (Fig 10A), colorimetric assays were carried out to determine the GTPase kinetics of 0.5 μM Dnm1, a concentration used in a previous study [38], and the control samples, including 5 μM and 10 μM His-Yta4(ΔTM) and the buffer used to purify His-Yta4(ΔTM) and Dnm1, following the method described previously (see details in the Methods section) [39,40]. Consistently, Dnm1, but not the control samples, exhibited a significant GTPase activity ($K_m$ = 193.80 μM, $V_{max}$ = 7.03 μM/min). The $K_m$ value determined by using fission yeast Dnm1 alone (at 0.5 μM) in the present study was approximately 2 times larger than the $K_m$ value determined previously using budding yeast Dnm1 (at 0.5 μM) [38] (Fig 10B). This could be due to the species-specific properties of Dnm1. Alternatively, since Dnm1 forms heterogeneous oligomers in vitro [37,38], the heterogeneity may contribute to the different $K_m$ values determined in each kinetics assay. Therefore, to precisely assess the effect of Yta4 on the GTPase activity of Dnm1, we performed parallel colorimetric assays using a single batch of recombinant Dnm1 and His-Yta4(ΔTM) (Fig 10C). In the presence of 5 μM and 10 μM His-Yta4(ΔTM), the $K_m$ value of Dnm1 was increased approximately 3 and approximately 6-fold (3 independent experiments), respectively, indicating that His-Yta4(ΔTM) reduced the affinity of Dnm1 for GTP. In contrast, the $V_{max}$ value was not significantly affected by His-Yta4(ΔTM) (3 independent experiments), suggesting that His-Yta4(ΔTM) did not affect the rate of Dnm1-mediated GTP hydrolysis (Fig 10C). Collectively, we concluded that Yta4 is a competitive inhibitor of Dnm1.

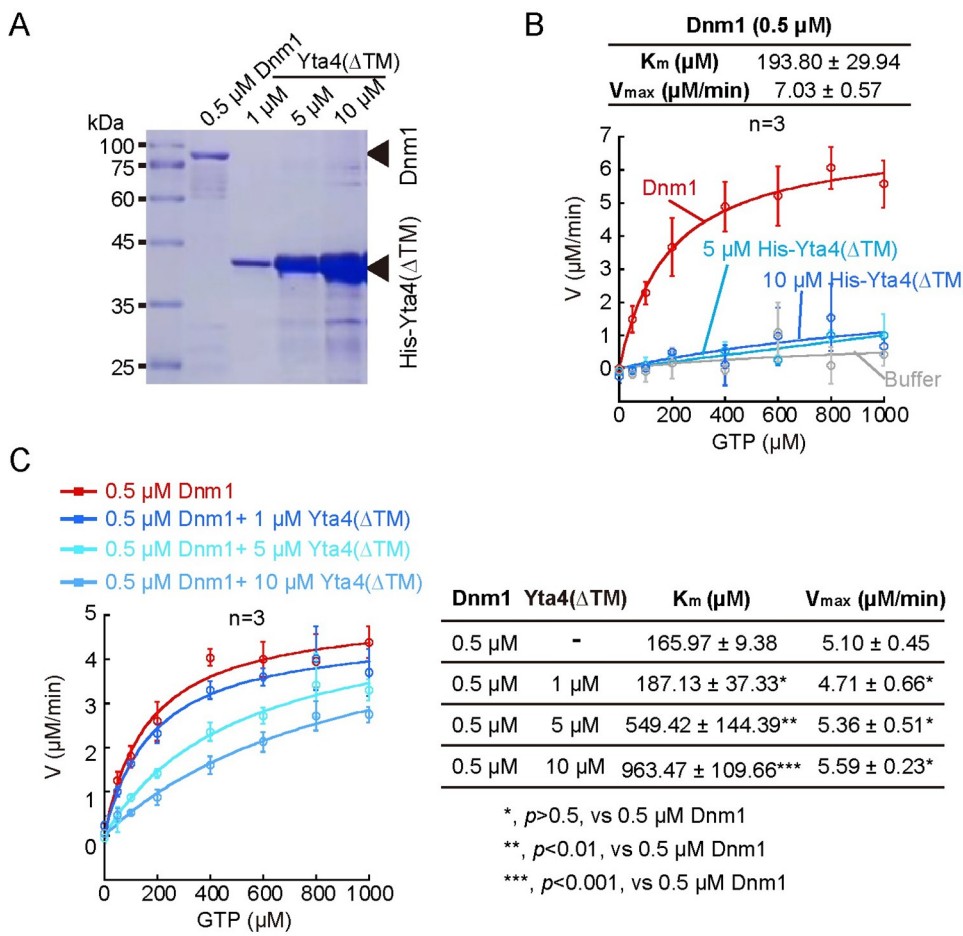

**Fig 10. Yta4 decreases the affinity of Dnm1 for GTP.** (A) A gel stained by Coomassie blue display the indicated proteins used in the GTPase kinetics experiments shown in (B) and (C). The concentrations of Dnm1 and His-Yta4 (ΔTM) are indicated. (B) GTPase kinetics assays were performed with the indicated proteins: 0.5 μM Dnm1 alone, 5 μM His-Yta4(ΔTM), and 10 μM His-Yta4(ΔTM). The buffer was used as a negative control. The initial rate (V) was determined using the slope of generated phosphate measured by colorimetric assays at 37°C at the indicated concentration of GTP. The curves were created by fitting to a Michaelis–Menten model, and $K_m$ and $V_{max}$ values were obtained from the fitting. Data points are averages, while error bars represent SD (from 3 independent experiments). Values in the table are average±SD. Raw data are available in S1 Data. (C) GTPase kinetics assays were performed with the indicated proteins: 0.5 μM Dnm1 alone, 0.5 μM Dnm1 plus 1 μM His-Yta4(ΔTM), 0.5 μM Dnm1 plus 5 μM His-Yta4(ΔTM), and 0.5 μM Dnm1 plus 10 μM His-Yta4(ΔTM). The curves were created by fitting to a Michaelis–Menten model, and $K_m$ and $V_{max}$ values were obtained from the fitting. Data points are averages, while error bars represent SD (from 3 independent experiments). Values in the table are average±SD. Student's *t* test was used to calculate *p* values. Raw data are available in S1 Data.

Dnm1 forms rings or spiral-like structures in the presence of 1 mM GMP-PCP, the nonhydrolyzable GTP analog [37]. Therefore, we tested the effect of Yta4 on Dnm1 assembly by centrifugation. First, we titrated the concentration of GMP-PCP and assessed the effect of GMP-PCP on Dnm1 assembly. As shown in Fig 11A and 11B, the fraction of Dnm1 (at 2 μM) in the pellet increased as the concentration of GMP-PCP increased from 0 μM to 100 μM, suggestive of the formation of higher-order Dnm1 structures in the presence of a higher concentration of GMP-CPP. Moreover, negative stain electron microscopic observation confirmed that Dnm1 (at 2 μM) mainly formed rings in the presence of 100 μM GMP-PCP but some Dnm1 spiral-like structures were also detected (Fig 11C). By contrast, in the absence of GMP-PCP, Dnm1 (at 2 μM) formed only small filament-like structures (Fig 11C). The

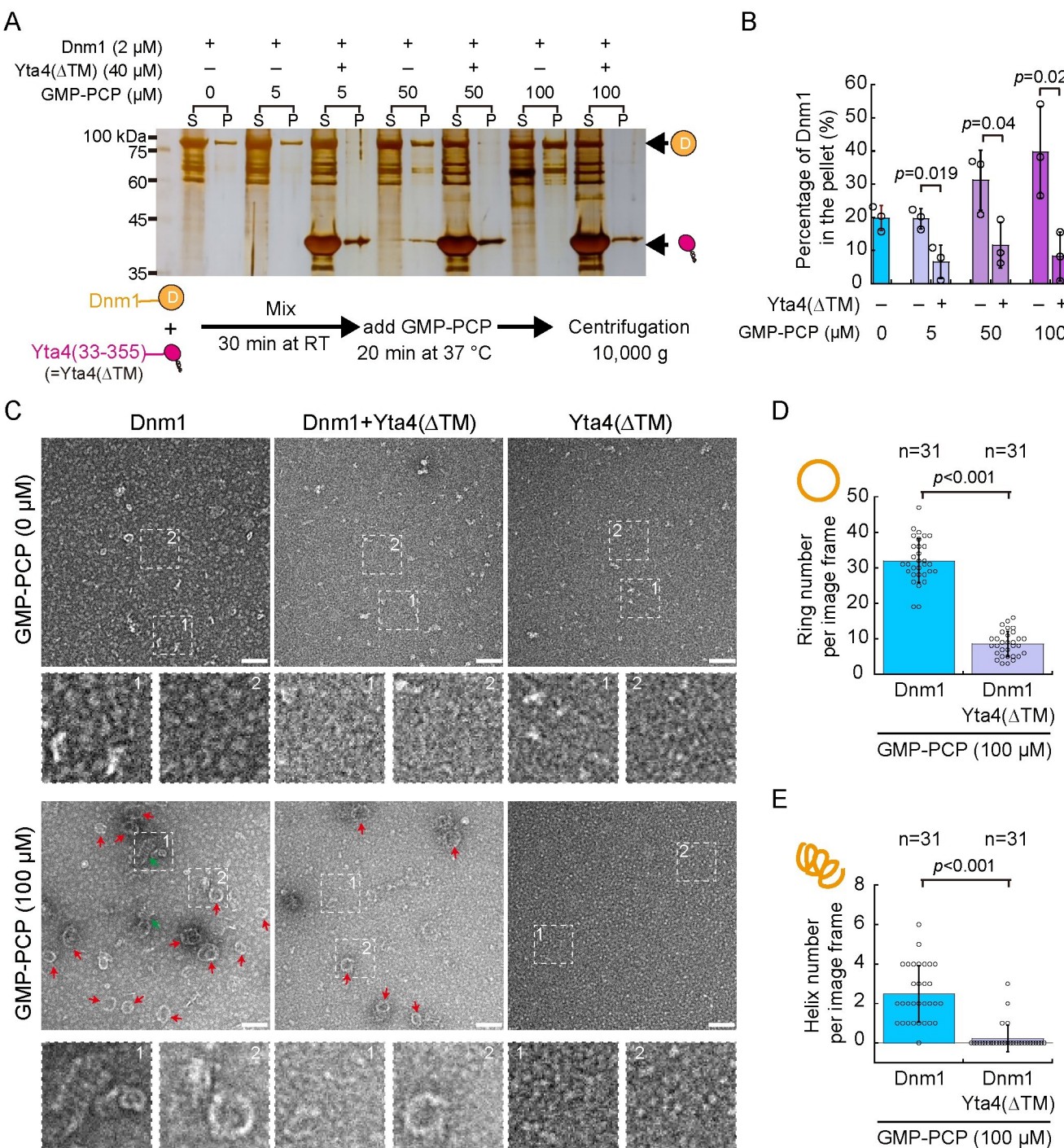

**Fig 11. Yta4 inhibits Dnm1 assembly.** (A) Velocity sedimentation assays. SDS–PAGE was performed to analyze the effect of His-Yta4(ΔTM) on Dnm1 assembly in the presence of the nonhydrolyzable GTP analog GMP-PCP (at 0, 5, 50, and 100 μM). A group of samples lacking His-Yta4(ΔTM) were included. *S* and *P* represent the supernatant and pellet of the indicated samples after centrifugation, respectively. Note that more Dnm1 was present in the pellet of the samples without His-Yta4(ΔTM) as the concentration of GMP-PCP increased but the presence of His-Yta4(ΔTM) significantly reduced the amount of Dnm1 in the pellet fractions. (B) Quantification of the percentage of Dnm1 in the pellet. The top of the column indicates the mean while error bars represent SD. Student's *t* test was used to calculate *p* values. Raw data are available in S1 Data. (C) Representative images of negative staining EM. Dnm1 alone, His-Yta4(ΔTM) alone, or His-Yta4(ΔTM) and Dnm1 in combination were incubated in buffers containing or lacking 100 μM GMP-PCP. Images were acquired at a magnification of 29,000. Red arrowheads mark Dnm1 rings while green arrows indicate spiral-like structures. Two numbered magnified images are indicated by the dashed squares. Scale bar, 100 nm. (D and E) Quantification of Dnm1 rings and spiral-like structures in the negative staining EM samples indicated in (C). Quantification was performed with images acquired at a magnification of 29,000. The number of image frames

analyzed is indicated (*n*). Statistical analysis was performed by Student's *t* test. GMP-PCP was added after the 2 proteins have been mixed. Note that Yta4 inhibits GMP-PCP induced Dnm1 oligomerization. Raw data are available in S1 Data. EM, electron microscopy.

presence of His-Yta4(ΔTM) (at 40 μM) significantly reduced the fraction of Dnm1 in the pellet (Fig 11A and 11B). Consistently, negative stain electron microscopic observation revealed that the presence of His-Yta4(ΔTM) significantly reduced the number of Dnm1 rings and spiral-like structures induced by GMP-PCP (Fig 11C–11E). Hence, these results show the characteristic property of Yta4 in reducing the affinity of Dnm1 for GTP and in inhibiting Dnm1 assembly.

GTP drives Dnm1 assembly, and Dnm1 assembly in turn stimulates GTP hydrolysis of Dnm1 [37,38]. Moreover, Dnm1 can self-assemble to form heterogenous oligomers in vitro [37,38]. These intertwining effects of Dnm1 make it challenging to clearly determine how Yta4 affects the GTPase activity of Dnm1. To reduce the complexity in analyzing Dnm1 GTPase activity, we performed colorimetric assays by using the assembly-defective version of Dnm1 (i.e., Dnm1(G380D), see S6A Fig). Interestingly, Dnm1(G380D) still exhibited a GTPase activity but has a very low rate of GTP hydrolysis ($K_m$ = 85.00 μM, $V_{max}$ = 1.02 μM/min, versus the control Dnm1(WT): $K_m$ = 126.59 μM, $V_{max}$ = 4.82 μM/min) (S8 Fig). Of note, the presence of His-Yta4(ΔTM) similarly increased the $K_m$ values, but not the $V_{max}$ values, of both Dnm1 (WT) and Dnm1(G380D) (S8 Fig). This result suggests that Yta4 can inhibit the GTPase activity of Dnm1 independent of Dnm1 self-assembly.

## Discussion

It has been established that ATAD1 family proteins safeguard mitochondrial quality by clearing mistargeted TA proteins on mitochondria [5]. However, how ATAD1 is involved in regulating mitochondrial dynamics has remained elusive. In the present study, we propose a model that Yta4 (the fission yeast homolog of ATAD1) inhibits mitochondrial fission by disrupting the mitochondrial fission machinery through regulating the mitochondrial divisome (Fig 12). This model is supported by the following lines of evidence.

First, the absence of Yta4 enhanced mitochondrial fission (Figs 1G, 3E and S3D) and resulted in mitochondrial fragmentation (Fig 1A and 1C). This is consistent with the previous finding that depletion of ATAD1 in mouse embryonic fibroblasts leads to mitochondrial fragmentation [4]. Intriguingly, the absence of Msp1 (the budding yeast homology of ATAD1) does not appear to cause mitochondrial fragmentation [3,4]. This may be due to the lack of interaction between Msp1 and the mitochondrial fission receptor Fis1 in budding yeast under physiological conditions [4,11] (see discussion below). Nonetheless, mitochondrial morphology is significantly altered in cells lacking both Msp1 and the GET system component Get1, Get2, or Get3 (i.e., the system mediating the insertion of newly synthesized TA proteins into the ER) [3,4]. Together, these findings indicate that Yta4/ATAD1 likely functions under physiological conditions to inhibit excessive mitochondrial fission.

Second, overexpression of Yta4 impaired the localization of Dnm1, Mdv1, and Fis1 to mitochondria (Figs 4D, 4E, 7A, S1C and S1D). Msp1/ATAD1 functions to clear mistargeted TA proteins on mitochondria [5], and Fis1 is a typical TA protein on the mitochondrial outer membrane [4]. Consistently, our data show that overexpression of Yta4 removed Fis1 from mitochondria (Figs 4E, 5C, 5D and S1D), and this depended on the hydrolysis and translocation activities of Yta4 because overexpression of the ATP hydrolysis-deficient mutant Yta4 (EQ) or the translocation-deficient mutant Yta4(AA) failed to delocalize Fis1 from mitochondria (Figs 4E and S1D). Given that Fis1 is responsible for recruiting Dnm1 to mitochondria via Caf4/Mdv1 [31–35], it is possible that the impaired mitochondrial localization of Dnm1 in

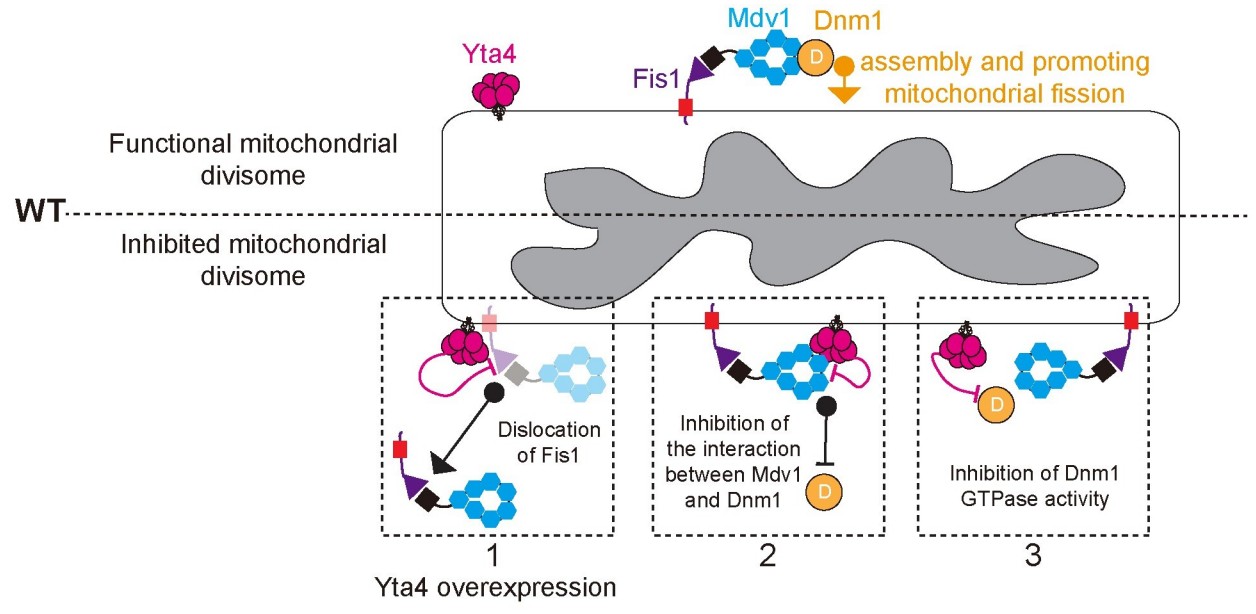

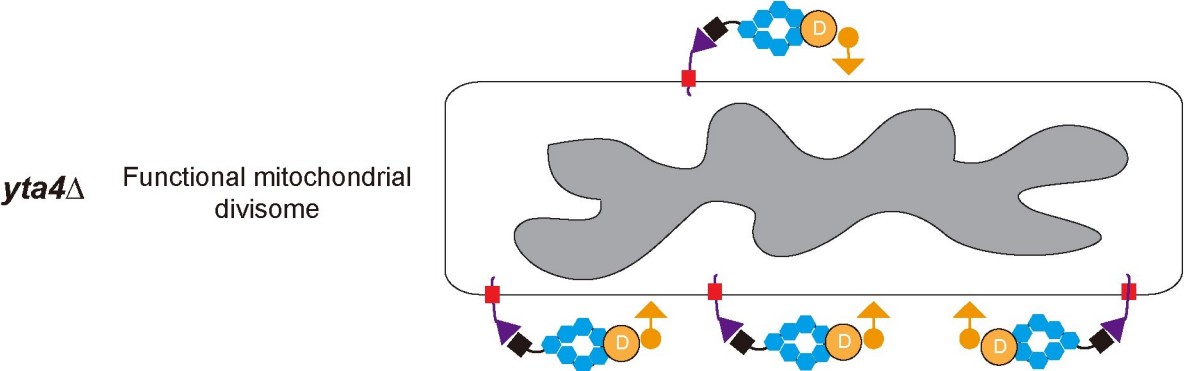

**Fig 12. Working model illustrating the role of Yta4 in inhibiting mitochondrial fission.** Mitochondrial fission is mediated by Dnm1, which is recruited to the mitochondrial outer membrane by Fis1 via Mdv1. Note that Fis1, Mdv1, and Dnm1 are the 3 components of the mitochondrial divisome [25]. Yta4 interacts with Fis1, Mdv1, and Dnm1 to inhibit mitochondrial fission by different mechanisms (top diagram). (1) Yta4 likely functions as a canonical dislocase to maintain the homeostasis of the TA protein Fis1 on the mitochondrial outer membrane, and the delocalization of Fis1 from mitochondria is significant when Yta4 is overexpressed, (2) Yta4 and Dnm1 interact with Mdv1 in a competitive manner to inhibit the recruitment of Dnm1, and (3) Yta4 inhibits the GTPase activity of Dnm1 likely through a dynamic interaction. Thus, the absence of Yta4 promotes efficient mitochondrial fission (bottom diagram). TA, tail-anchored; WT, wild-type.

Yta4-overexpressing cells is a consequence of the lack of Fis1 and Mdv1 on mitochondria. Our data also showed that overexpression of the ATP hydrolysis-deficient mutant Yta4(EQ) or the translocation-deficient mutant Yta4(AA) failed to delocalize Fis1 and Mdv1 from mitochondria but impaired the localization of Dnm1 to mitochondria (Figs 4D, 4E, 7A, S1C, and S1D). Therefore, it is also possible that Yta4 directly inhibit the fission capability of Dnm1 in an enzymatic activity-independent manner (Fig 12, dashed squares 2 and 3).

Third, Yta4 physically interacts with Dnm1 (Fig 8B), reduces the affinity of Dnm1 for GTP in a competitive manner (Fig 10C), and inhibits Dnm1 assembly in vitro (Fig 11). In addition, Yta4 physically interacts with Mdv1 (Fig 8D) and Fis1 (Fig 8C), which are responsible for recruiting Dnm1 to mitochondria. Surprisingly, in budding yeast, Msp1 (Yta4 homolog) does not interact with Fis1, but insertion of a Pex15 hydrophobic patch into the region near the

transmembrane domain of Fis1 enables Fis1 to become a client protein of Msp1 [11]. The lack of interaction between Fis1 and Msp1 likely explains why the absence of Msp1 alone does not affect mitochondrial morphology in budding yeast [3,4]. By contrast, in fission yeast, Yta4 physically interacts with Fis1, Mdv1, and Dnm1, the 3 components of the mitochondrial divisome. This finding indicates that the specificity of recognition of client proteins by Yta4 may be low because Fis1 (a TA protein), Mdv1 (a non-TA protein), and Dnm1 (a non-TA protein) are different types of proteins and multiple TA (Pex15, Gos1, Fmp32, Frt1, and Ysy6) and non-TA proteins (AMPAR, Cis1, PLAA, and UBXN4) have been shown to be client proteins of the homologs of Yta4, i.e., Msp1 and ATAD1. Although the expression levels of Dnm1 were comparable between WT and *yta4Δ* cells (Fig 3B), the percentage of Dnm1 foci associated with mitochondrial fission increased significantly in *yta4Δ* cells (Fig 3E) and no mitochondrial fragmentation was detected in cells lacking both Yta4 and Dnm1 (Fig 3A). Hence, Yta4 functions to inhibit Dnm1 to prevent excessive mitochondrial fission. In evolution, the fission yeast *Schizosaccharomyces pombe* exhibits more ancient traits than the budding yeast *S. cerevisiae* [41]. Therefore, it is conceivable that the interplay between Yta4 and the mitochondrial divisome may represent an ancient but conserved regulatory mechanism of the AAA-ATPase in regulating mitochondrial dynamics.

Why does Yta4 interact with the 3 components of the mitochondrial divisome (i.e., Fis1, Mdv1, and Dnm1) to regulate mitochondrial fission? We speculate that the interaction between Yta4 and Fis1 may maintain the homeostasis of Fis1 on mitochondria (Fig 12, dashed square 1) so that the mitochondrial sites for the assembly of the mitochondrial divisome can be spatially regulated. However, we noticed that the absence of Yta4 did not significantly alter the localization of Fis1 on mitochondria (Figs 4E and S3E). Considering that Yta4 have many client proteins, including Mdv1 and Dnm1, on the mitochondrial outer membrane, we speculate that the minor effect of the absence of Yta4 on Fis1 could be due to the infrequent extraction of Fis1 by Yta4 under physiological conditions. In addition, within the cell, Yta4 may not be present as only hexamers, a form required for extracting TA proteins. The interactions of Yta4 with Mdv1 and Dnm1 have a synergistic effect on inhibiting mitochondrial fission. First, the interaction between Yta4 and Mdv1 blocks the formation of the Mdv1-Dnm1 complex (Fig 9A), which may directly prevent the recruitment of Dnm1 to mitochondria (Fig 12, dashed square 2). Second, our GTPase kinetics assays clearly demonstrated that Yta4 reduces the affinity of Dnm1 for GTP in a competitive manner (Fig 10C). Considering the fact that Mdv1 preferentially interacts with GTP-bound Dnm1 and promotes Dnm1 assembly [38], we speculate that the interaction between Yta4 and Dnm1 is dynamic but the dynamic interaction is sufficient to reduce the amount of GTP-bound Dnm1, thus inhibiting Dnm1 function by preventing the recruitment of Dnm1 by Mdv1 (Fig 12, dashed square 3). Therefore, the interaction of Yta4 with the 3 components of the mitochondrial divisome may simply ensure efficient inhibition of mitochondrial fission.

Multiple mechanisms underlying suppression of mitochondrial fission have been established. Most of these mechanisms involve posttranslational modifications of Dnm1 (Drp1 in mammals) or its mitochondrial adaptor proteins, e.g., Fis1, Mff, MiD49, and MiD51 [26]. These posttranslational modifications usually take place in response to internal or external stimuli and the consequent changes of mitochondrial morphology endow the cell with more adaptability for altered functional demands. However, the inhibitory effect of Yta4 on mitochondrial fission we discovered here does not appear to depend on posttranslational modifications but arises from a distinct mechanism that relies on protein–protein interactions. This new mechanism takes effect constitutively to prevent excessive mitochondrial fission and acts as a safeguard for the integrity of the mitochondrial network, especially under stress conditions

when cells are prone to mitochondrial fragmentation, such as with enhanced mitochondrial respiration or depolarized mitochondria.

In conclusion, this study reveals an uncharacterized role of the conserved AAA-ATPase Yta4/ATAD1 in inhibiting excessive mitochondrial fission, which may be essential to maintain the fission-fusion balance of mitochondria under both normal and pathological conditions.

## Materials and methods

### Molecular cloning and yeast strain construction

The plasmids used in this study (S1 Table) were constructed by the conventional method of digestion and ligation with restriction enzymes (NEB) and T4 ligase (NEB) or by recombination with Exanase II (Vazymebiotech). The ClonExpress II One Step Cloning Kit (Vazymebiotech) was used to generate different point mutations of Yta4.

Yeast strains (S2 Table) were constructed by random spore analysis or tetrad dissection (Singer instruments). Gene deletion and tagging were achieved by the PCR-based homologous recombination method [42]. Plasmids were transformed into yeast cells by the lithium acetate method. Unless specified otherwise, cells were grown in EMM medium (Edinburgh minimal media) containing the 5 supplements (0.225 g/L each) (Formedium): adenine, leucine, uracil, histidine, and lysine (referred to as EMM5S).

### Profusion chamber and FCCP treatment experiments

For coverslip coating, 5 mg/ml polylysine (Sigma-Aldrich) was used. To create a profusion chamber, 2 pieces of double sticky tape were used to attach the polylysine-coated coverslip to a slide [43]. The profusion chambers were washed with 100 µL of EMM5S before the cells were injected. After injection of yeast cells, the chamber was placed upside down to allow attachment of the yeast cells to the polylysine-coated coverslip at 30°C for 10 min. Unattached cells were washed away with 100 µL of EMM5S. To induce mitochondrial depolarization, 80 µL of 0.5 µM or 2 µM FCCP (carbonyl cyanide 4-(trifluoromethoxy) phenylhydrazone) (Sigma-Aldrich) prepared in a glass tube was used to treat cells in a profusion chamber. Time-lapse imaging was conducted to observe morphological changes in mitochondria, and images were acquired immediately after the replacement of EMM5S containing 0.5 µM or 2 µM FCCP.

### Microscopy and data analysis

The fission yeast strains used for imaging were cultured in EMM5S medium at 30°C. Imaging was carried out with a PerkinElmer UltraVIEW VoX spinning disk confocal microscope equipped with a Hamamatsu C9100-23B electron multiplying charge-coupled device (EMCCD) camera and a Nikon CFI Apochromat TIRF 100× objective (NA = 1.49). FCCP treatment was performed in profusion chambers, while regular imaging was performed on EMM5S–agarose (3%) pad slides [44]. For time-lapse imaging, we acquired stack images containing 11 planes (0.5 µm/5 µm) every 30 s or 1 min. For maximum projection images, we acquired stack images containing 11 planes (0.5 µm/5 µm). All images were acquired with Volocity (Perkinelmer).

Imaging data were analyzed with MetaMorph (Moleculardevices) and Fiji ImageJ (NIH). The mitochondrial number was quantified with the algorithm MiNA [45]. Graphs and plots were generated using KaleidaGraph (version 4.5) (Synergy). Statistical analysis was performed with Microsoft Excel and/or KaleidaGraph (version 4.5). The normality of the data was determined by OriginPro (version 2021b) using the Shapiro–Wilk test. $p$ Values were calculated with the statistical tests specified in the figure legends.

## Isolation of mitochondria

To isolate mitochondria, fission yeast strains were inoculated in EMM5S medium. Exponentially grown cells were harvested from 20 ml culture and the mitochondrial fraction and the cytosolic fraction were separated by the Mitochondrial Yeast Isolation Kit (Abcam). Whole cell lysate (W), the mitochondrial fraction (M), and the cytosolic fraction (C) were analyzed by SDS-PAGE and western blotting with antibodies against GFP (dilution: 1:2,000, Rockland, 600-101-215), Dnm1 (1:2,000, Homemade), Mti3(1:2,000, Homemade), and Tubulin (1:10,000, Bioacademia, 63–160).

## Protein expression and purification

His-Yta4(a.a. 33–355) (lacking the N-terminal transmembrane domain), Hcp1-Yta4(a.a. 35–355), Hcp1-Msp1(a.a. 36–362) (lacking the N-terminal transmembrane domain), and His-Fis1 (a.a. 1–128) (lacking the C-terminal transmembrane domain) were purified from the *E. coli* BL21, while Dnm1 was purified from the *E. coli* Rosseta, using Ni-NTA resins (QIAGEN). GST, GST-Hcp1, GST-Yta4(a.a. 33–355), GST-Hcp1-Yta4(a.a. 35–355), GST-Mdv1(a.a. 319–651), and Fis1(a.a. 1–128)-GST co-expressed with His-Mdv1(a.a. 1–249) were purified from the *E. coli* BL21 using Glutathione Sepharose 4B resins (GE healthcare).

His-Yta4(a.a. 33–355), His-Fis1(a.a. 1–128), and His-sumo-Dnm1 and His-sumo-Dnm1 (G380D) were expressed in *E. coli* (Dnm1 and Dnm1(G380D) in Rosseta and others in BL21), and 0.5 mM IPTG was used for induction at 16˚C overnight. The His-fused proteins were purified with Ni-NTA resins (QIAGEN) using Lysis/Wash buffer (25 mM HEPES, 500 mM NaCl, 30 mM imidazole, 5 mM $MgCl_2$, 10 mM β-Mercaptoethanol (pH 7.5)) and Elution buffer (25 mM HEPES, 500 mM NaCl, 500 mM imidazole, 5 mM $MgCl_2$, 10 mM β-Mercaptoethanol (pH 7.5)). His-sumo-Hcp1-Yta4(a.a. 35–355) and His-sumo-Hcp1-Msp1(a.a 36–362) expressed in the *E. coli* BL21 were purified as previously described [10]. After elution, His-sumo-Dnm1, His-sumo-Yta4(a.a. 33–355), and His-sumo-Hcp1-Msp1(a.a. 36–362) were digested with a sumo protease to remove the tag His-sumo. For in vitro biochemistry assays, enzymic kinetics assays, and cross-linking assays, His-Yta4(a.a. 33–355), Hcp1-Yta4(a.a. 35–355), His-Fis1(a.a. 1–128), and Dnm1 were then further purified using an AKTA prime system with SEC buffer (25 mM HEPES, 150 mM KCl, 10 mM β-Mercaptoethanol, 5 mM $MgCl_2$ (pH 7.5)). In addition, Hcp1-Msp1(a.a. 36–362) was further purified with SEC buffer as previously described [10]. Glycerol was then added into the concentrated proteins, using a 30 MWCO concentrator, to a final concentration of 5% as the cryoprotectant. Finally, the aliquots were frozen in liquid nitrogen and stored at − 80˚C.

GST, GST-Hcp1, GST-Yta4(a.a. 33–355), and GST-Hcp1-Yta4(a.a. 35–355) were purified with Glutathione Sepharose 4B resins using 1 × TBST buffer (Tris-buffered saline buffer plus 0.1% Triton X-100). GST-Mdv1(a.a. 319–651) and Fis1(a.a. 1–128)-GST co-expressed with His-Mdv1(a.a. 1–249) were induced for expression with 0.5 mM IPTG at 22˚C overnight, and purified using phosphate buffer (50 mM $Na_3PO_4$, 150 mM NaCl, 10 mM imidazole plus 0.1% Triton X-100). Glycerol was then added to a final concentration of 20% as the cryoprotectant. Finally, the aliquots resins were stored at −20˚C.

## Biochemistry assays

For analysis of protein expression levels, protein extract was prepared by the NaOH lysis method [46]. Exponentially grown cells harvested from 6 ml culture were resuspended in 0.5 ml ddH$_2$O after washed once with 1 ml distilled deionized water (ddH$_2$O), and then 0.5 ml NaOH (0.6 M) was added. The suspension was mixed gently and incubated at room temperature for 10 min. Cell pellets were collected by centrifugation at 6,000 rpm for 1 min. The cell

pellets were boiled in SDS sample buffer (60 mM Tris-HCl (pH 6.8), 4% SDS, 4% β-mercaptethanol, 5% glycerol, and 0.002% bromophenol blue) for 5 min and analyzed by SDS-PAGE and western blotting with antibodies against GFP (1:2,000, Rockland, 600101215), Dnm1 (1:2,000, Homemade), Myc (1:2,000, Thermo Scientific, SK260674), tdTomato (1:2,000, Homemade), and Tubulin (1:10,000, Bioacademia, 63–160).

For Co-IP assays, fission yeast strains were inoculated in EMM5S medium. Exponentially grown cells were harvested from 500 ml culture and were ground in liquid nitrogen with the mortar grinder RM 200 (Retsch). After grinding, cells were dissolved in 1 × TBST buffer (Tris-buffered saline buffer plus 0.5% Triton X-100) supplemented with cocktail protease inhibitors and 1 mM PMSF at 4˚C for 30 min by gently rotating. Cell lysates were then centrifuged at 12,000 rpm at 4˚C for 30 min, and supernatants were collected as input samples and for Co-IP. Briefly, Dynabeads protein G beads (Thermo Scientific) bound with antibodies against GFP (Homemade) or Dnm1 (Homemade) were added to the supernatants and the mixture was incubated at 4˚C on a rotator for 2 h. The Dynabeads protein G beads were then washed with 1 × TBST (Tris-buffered saline buffer plus 0.1% Triton X-100) buffer for 5 times and 1 × TBS for 1 time. Then, the input samples and beads were boiled in SDS sample buffer (60 mM Tris-HCl (pH 6.8), 4% SDS, 4% β-mercaptethanol, 5% glycerol, and 0.002% bromophenol blue) at 100˚C for 5 min. Co-IP protein samples were analyzed by SDS-PAGE and western blotting with antibodies against GFP (1:2,000, Rockland, 600101215), Dnm1 (1:2,000, Homemade), and Myc (1:2,000, Thermo Scientific, SK260674).

## GST pull-down assays

For GST pull-down assays, the baits of GST-fused proteins were purified with Glutathione Sepharose 4B resins, and the preys of His-fused proteins were purified with Ni-NTA resins. In the assays, empty glutathione resins and glutathione resins bound with GST were used as negative controls. His-fused proteins were precleaned with glutathione resins at 4˚C for 10 min. The baits and preys were then incubated together in TBST buffer (Tris-buffered saline buffer plus 0.5% Triton X-100) with 5% glycerol at 4˚C for 2 h.

To test the interaction of Yta4 monomers or hexamers with Dnm1 or Fis1, glutathione resins bound with GST-Yta4(a.a. 33–355) or GST-Hcp1-Yta4(a.a. 35–355) were incubated with Dnm1 or His-Fis1(a.a. 1–128), respectively. To test the interaction of Mdv1 with Yta4 or Dnm1, glutathione resins bound with GST-Mdv1(a.a. 319–651) were incubated with His-Yta4 (a.a. 33–355) or Dnm1, respectively. Note that when His-Fis1(a.a. 1–128) was used as the preys, imidazole was added into TBST buffer to a final concentration of 40 mM, and to test the interaction of Fis1 with Yta4 hexamers, the buffer contained 0 or1 mM ATP.

To test the effect of Yta4 monomers on the interactions of Mdv1 with Dnm1/Dnm1 (G380D) and Fis1, glutathione resins bound with GST-Mdv1(a.a. 319–651) or glutathione resins bound with GST-Mdv1(a.a. 1–249) were incubated with Dnm1/Dnm1(G380D) or His-Fis1(a.a. 1–128), respectively, with or without His-Yta4(a.a 33–355). To test the effect of Yta4 hexamers on the interaction of Mdv1 with Fis1, glutathione resins bound with the co-expressed Fis1(a.a. 1–128)-GST and His-Mdv1(a.a. 1–249) were incubated with or without Hcp1-Yta4(a.a. 35–355) in the presence of 5 mM ATP.

After incubation, glutathione resins were washed with TBST buffer for 5 times and the TBS buffer for once, followed by boiling in SDS sample buffer at 100˚C for 5 min. The pull-down protein samples were then analyzed by SDS-PAGE and western blotting with antibodies against His (1:2,000, Abclonal, AE003), Dnm1 (1:2,000, Homemade), Yta4 (1:2,000, Homemade), and GST (1:2,000, Abclonal, AE006).

Secondary antibodies used in this study are as follows: HRP Goat Anti-Mouse Conjugate (1:10,000, Abclonal, AS003), Goat Anti-Rabbit-HRP Conjugate (1:10,000, Bio-rad, 170–5046), and Rabbit Anti-Goat HRP Conjugate (1:10,000, Abclonal, AS029).

## ATPase and GTPase kinetics assays

For GTPase kinetics assays, 50 μL 2.4 × Dnm1/Dnm1(G380D) and 30 μL 4 × His-Yta4 or a storage buffer (25 mM HEPES, 150 mM KCl, 10 mM β-mercaptoethanol, 5 mM $MgCl_2$, and 5% Glycerol; pH = 7.5) were first mixed in PCR tubes for 30 min at room temperature, and 40 μL 3 × GTP/$MgCl_2$, diluted with Assembly buffer (25 mM HEPES, 150 mM KCl, 10 mM β-mercaptoethanol, pH = 7.5) was then added to the mixture. The final concentration of Dnm1/Dnm1(G380D) was 0.5 μM and the final concentrations of His-Yta4(a.a. 33–355) were 1, 5, and 10 μM, respectively.

For ATPase kinetics assays, 30 μL 4 × His-Yta4 (a.a. 33–355) or Hcp1-Yta4 (a.a. 35–355) were mixed with 50 μL Assembly buffer, and 40 μL 3 × ATP/$MgCl_2$ was then added into the mixture. The final concentration of His-Yta4 (a.a. 33–355) and Hcp1-Yta4 (a.a. 35–355) was 1 μM.

In the reaction mixture, the concentration of ATP or GTP ranged from 0 μM to 1,000 μM (0, 50, 100, 200, 400, 600, and 1,000 μM, respectively). Upon the addition of GTP or ATP, PCR tubes were moved to a heat block at 37°C to start the ATP or GTP hydrolysis reaction.

At the time points 0, 2.5, 5, 7.5, 10, and 20 min, 18 μL of the mixture was quickly taken to the wells containing 0.5 M EDTA to stop the reaction. Finally, 135 μL Malachite Green Reagent (1 mM Malachite Green Carbinol hydrochloride, 10 mM ammonium molybdate, and 1 N HCl) was added to each well, and the plates were read at A650 on a Synergy H1 Microplate reader (BioTek). The slope of the generated phosphate measured by the colorimetric assay was used to determine the initial rate of ATP and GTP hydrolysis, and the initial rates were then plotted against the concentration of ATP or GTP. Finally, $K_m$ and $V_{max}$ were determined by fitting the plots with the Michaelis–Menten model using KaleidaGraph (Synergy).

## Velocity sedimentation

For the velocity sedimentation assays, 5 μL 2 × Dnm1 and 5 μL 2 × His-Yta4(a.a. 35–355) or 5 μL storage buffer (25 mM HEPES, 150 mM KCl, 10 mM β-mercaptoethanol, 5 mM $MgCl_2$, and 5% Glycerol; pH = 7.5) were first mixed in Eppendorf tubes for 30 min at room temperature, and 0.8 μL 25 × GMP-PCP/$MgCl_2$ was then added to the mixture. The final concentrations of Dnm1 and His-Yta4(a.a. 33–355) were 2 μM and 40 μM, respectively, and the final concentration of GMP-PCP ranged from 0 μM to 100 μM (i.e., 0, 5, 50, and 100 μM). After incubation for 20 min at 37°C on a heat block, the mixtures were centrifuged at 10,000 g for 15 min at 4°C. The supernatant (S) was collected in tubes, and the pellet (P) was resuspended in assembly buffer. Finally, the supernatant and pellet fractions were analyzed by SDS-PAGE and silver staining. To quantify the percentage of Dnm1 in each fraction to the sum, the band intensity was measured by MetaMorph (Molecular Devices), and calculation was performed with Excel.

## Negative staining electron microscopy

To prepare negative-stain EM specimens, 5 μL 2 × Dnm1 and 5 μL 2 × His-Yta4 (a.a. 33–355) or a storage buffer were first mixed for 30 min at room temperature, and 0.4 μL 25 × GMP-PCP and 0.4 μL 25 × $MgCl_2$ were then added into the mixture. The final concentrations for Dnm1 and His-Yta4 (a.a. 33–355) were 2 μM and 40 μM, respectively, and the final concentration of GMP-PCP is 100 μM. After incubation for 20 min at 37°C on a heat block, 3 μL mixture was

dropped on a carbon-coated grid, and after 30 s, the grid was blotted with a filter paper, stained with 2% uranyl acetate for 90 s, blotted again, and air dried. All samples were examined with a Tecnai G2 twin electron microscope (FEI). Images were acquired at a magnification of 29,000× or 62,000×. Images were recorded digitally on a 4,000 × 4,000 CCD detector (FEI Eagle), operating at an acceleration voltage of 200 kV. Quantification was done by manually counting Dnm1 rings and filaments on each image frame.

## Size-exclusion chromatography

To analyze the molecular size of Hcp1-Yta4(a.a. 35–355), size-exclusion chromatography was performed with hypersaline buffer (50 mM HEPES, 500 mM NaCl, 500 mM KCl, 10 mM $MgCl_2$, 100 mM imidazole, 5% glycerol, 1 mM ATP, 0.5 mM EDTA (pH 7.5)) using the column Superdex 200 Increase 10/300 GL (GE, Cytiva) after a crude purification using the AKTA prime system. Hcp1-Msp1(ΔTM) was shown to be a hexamer [10]. Therefore, as a control, Hcp1-Msp1(a.a. 36–362) was analyzed following the same procedure above.

To analyze the molecular size of Dnm1 and Dnm1(G380D), size-exclusion chromatography was performed with SEC buffer (25 mM HEPES, 150 mM KCl, 10 mM β-Mercaptoethanol, 5 mM $MgCl_2$ (pH 7.5)) using the column Superdex 200 Increase 10/300 GL (GE, Cytiva).

## Cross-linking assays

His-Yta4(a.a. 33–355), Hcp1-Yta4 (a.a. 35–355), and Hcp1-Msp1(a.a. 36–362) were first diluted to a final concentration of 300 ng/μL using the corresponding buffer for purification. The cross-linking reagent EGS (Sigma-Aldrich) was added to the diluted samples at a final concentration of 4 mM. After incubation for 30 min at room temperature, the reaction was stopped by adding 10% of This-Glycine buffer (1 M Tris and 1 M Glycine (pH 7.5)) and analyzed by SDS-PAGE with Tris–Acetate polyacrylamide gradient gels (3% to 15%). In the analysis, HiMark pre-stained protein standard (Thermo Scientific) was used, and the gel were stained by silver staining. DMSO was used in parallel as negative controls.

## Supporting information

**S1 Fig. The effect of Yta4 overexpression on the localization of Dnm1 and Fis1 to mitochondria (related to Fig 4).** (A) Diagram illustrating the experimental procedure. Briefly, cells were precultured in EMM5S medium containing 0.3 μM thiamine, a chemical used to suppress the promoter *nmt41*; cells at the exponential phase were collected, washed, and cultured in thiamine-free EMM5S medium to allow expression of Yta4-13Myc (indicated by the yellow triangle) from the *nmt41* promoter. After culture in the thiamine-free EMM5S medium for 3, 6, 9, and 32 h, the cells were collected for microscopic observation and analysis by western blotting. (B) Testing the expression of Yta4(WT)/(AA)/(EQ)-13Myc (ectopically expressed from the *nmt41* promoter) in *yta4Δ* cells cultured in EMM5S medium containing 0.3 μM thiamine or thiamine-free EMM5S medium for the indicated time. Western blotting was performed with antibodies against Myc and Tubulin. (C) Maximum projection images of Dnm1-GFP-expressing *yta4Δ* cells that carry the indicated variants of Yta4-13Myc (from the *nmt41* promoter). Cells were cultured in EMM5S medium containing 0.3 μM thiamine or thiamine-free EMM5S medium for the indicated time, and mitochondria were stained with MitoTracker Red. Scale bars, 10 μm. (D) Maximum projection images of GFP-Fis1-expressing *yta4Δ* cells that carry the indicated variants of Yta4-13Myc (from the *nmt41* promoter). Cells were cultured in EMM5S medium containing 0.3 μM thiamine or thiamine-free EMM5S medium for the indicated time, and mitochondria were stained with MitoTracker Red. Scale bars, 10 μm. (TIF)

**S2 Fig. The localization of GFP-Fis1 and Ost4-tdTomato (an ER marker) in the indicated cells (related to Fig 4).** Maximum projection images of the indicated cells expressing GFP-Fis1 and Ost4-tdTomato (an ER marker). Cells were cultured in thiamine-free EMM5S medium for 20 h. Note that mitochondria were aggregated in cells expressing Yta4(AA) and Yta4(EQ), as shown in S1C and S1D Fig. Overexpression of Yta4 caused delocalization of GFP-Fis1 from mitochondria, and delocalized GFP-Fis1 likely localized within the cytoplasm, which did not colocalized with the ER marked by Ost4-tdTomato. Scale bars, 10 μm.
(TIF)

**S3 Fig. The effect of Yta4 and its mutants, expressed at the endogenous level, on mitochondria and on the localization of GFP-Fis1 and Dnm1-GFP (related to Fig 4).** (A) Maximum projection images of the indicated cells expressing GFP-Fis1 and Yta4(WT)/Yta4(AA)/Yta4(EQ)-13Myc (from the *yta4* promoter). Mitochondria were stained with MitoTracker Red. Note that the absence of Yta4 caused mitochondrial fragmentation and the expression of Yta4(WT)/Yta4(AA)/Yta4(EQ)-13Myc rescued the mitochondrial phenotype caused by the absence of Yta4. Scale bars, 10 μm. (B) Testing the expression of GFP-Fis1 and different Yta4-13Myc variants (expressed ectopically at the endogenous level) in the indicated cells used in (A) by western blotting with antibodies against GFP, Myc, and Tubulin. Note that the expression levels of Yta4(WT)-13Myc and Yta4(EQ)-13Myc were comparable but the expression level of Yta4(AA)-13Myc was relatively less. (C) Maximum projection images of the indicated cells expressing Dnm1-GFP and Yta4(WT)/Yta4(AA)/Yta4(EQ)-13Myc (from the *yta4* promoter). Mitochondria were stained with MitoTracker Red. Note that the absence of Yta4 consistently caused mitochondrial fragmentation and the expression of Yta4(WT)/ Yta4(AA)/ Yta4(EQ)-13Myc rescued the mitochondrial phenotype caused by the absence of Yta4. Scale bar, 10 μm. (D) Quantification of the percentage of Dnm1-associated mitochondrial fission. Statistical analysis was performed by Student's *t* test. The top of the column indicates the mean while bars indicate SD. The number of cells analyzed is shown on the x-axis. Experiments were repeated twice, and raw data are available in S1 Data. (E) Quantification of the average intensity of GFP-Fis1 signals on mitochondria in WT and *yta4Δ* cells. Cell number is indicated, and *a.u.* means arbitrary unit. Statistical analysis was performed by the Wilcoxon–Mann–Whitney rank sum test. Raw data are available in S1 Data.
(TIF)

**S4 Fig. Functional analysis of Yta4 monomers and hexamers (related to Fig 9).** (A) The recombinant proteins His-Yta4(ΔTM), Hcp1-Yta4(ΔTM), and Hcp1-Msp1(ΔTM) used in the analysis of size-exclusion chromatography (shown in B and C). (B and C) Size-exclusion chromatography profiles of the indicated proteins in (A). Protein standards are indicated. Raw data are available in S1 Data. (D) Cross-linking assays. Red, green, and black arrows indicate hexamers, dimers, and monomers, respectively. The proteins were treated with the cross-linking reagent EGS ("+") or DMSO ("-"). (E) ATPase kinetics assays were performed with the indicated proteins: 1 μM Hcp1-Yta4(ΔTM) and 1 μM His-Yta4(ΔTM). The buffer was used as a negative control. The initial rate (V) was determined using the slope of generated phosphate measured by colorimetric assays at 37˚C at the indicated concentration of ATP. The curves were created by fitting to a Michaelis–Menten model, and $K_m$ and $V_{max}$ values were obtained from the fitting. Data points are averages, while error bars represent SD (from 3 independent experiments). Values in the table are average±SD. Raw data are available in S1 Data.
(TIF)

**S5 Fig. The effect of Yta4 hexamerization on the interaction of Yta4 with Dnm1 and Fis1 (related to Fig 8).** (A) GST pull-down assays were performed to test the interaction between Dnm1 and GST-Hcp1-Yta4(ΔTM), GST-Yta4(ΔTM), GST-Hcp1, or GST. The GST-fused proteins were indicated by arrows. Western blotting was performed with antibodies against Dnm1 and GST. (B) GST pull-down assays were performed to test the interaction between His-Fis1 (ΔTM) and GST-Hcp1-Yta4(ΔTM), GST-Yta4(ΔTM), GST-Hcp1, or GST in the presence or absence of 1 mM ATP. Western blotting was performed with antibodies against Fis1 and GST. (TIF)

**S6 Fig. The effect of Dnm1 oligomerization on the interaction of Dnm1 with Mdv1 (related to Fig 9A).** (A) Top: Alignment of the amino acid sequences of Dnm1 and its budding yeast homolog. The conserved residues are highlighted in red, and the residues mutated (G was mutated into D) in Dnm1 are indicated by arrows. Bottom: Size-exclusion chromatography profiles of the indicated proteins. (B) GST pull-down assays were performed to test the interactions between GST-Mdv1(WD) and Dnm1(WT) and Dnm1(G380D) in the presence or absence of His-Yta4(ΔTM). GST was used as a negative control. Note that the presence of His-Yta4(ΔTM) significantly reduced the precipitate of Dnm1(WT) and Dnm1(G380D) by GST-Mdv1(WD). In addition, the precipitate of Dnm1(WT) and Dnm1(G380D) by GST-Mdv1(WD) was comparable. (C) Quantification of Dnm1 pulled down by GST-fused proteins. The band intensity of Dnm1 was first normalized to the band intensity of corresponding GST proteins, and the ratio of the indicated samples on x-axis of Dnm1 intensity was then calculated by normalizing to sample 2. Four independent experiments were performed. The top of the column represents the mean (indicated by *ave*), and single group Student's *t* test was used to calculate the *p* values. Raw data are available in S1 Data. (TIF)

**S7 Fig. Testing the competitive effect of Yta4 on the interaction of Mdv1 with Fis1 (related to Fig 9C).** (A) GST pull-down assays were performed to test the interaction between GST-Mdv1(NT) and His-Fis1(ΔTM) in the presence or absence of His-Yta4(ΔTM). Different concentration ratios (i.e., 1:1, 1:5, and 1:10) of Mdv1 to Yta4 were tested (see the input of His-Yta4(ΔTM) at the bottom left of the graph). GST was used as a negative control. Note that the presence of His-Yta4(ΔTM) did not significantly affect the interaction between GST-Mdv1 (NT) and His-Fis1(ΔTM). (B) Quantification of His-Fis1(ΔTM) pulled down by GST-fused proteins. The band intensity of His-Fis1(ΔTM) was first normalized to the band intensity of corresponding GST proteins, and the ratio of the normalized values of His-Fis1(ΔTM) intensity was then calculated. Yta4(-) and Yta4(+) indicate the samples, in which GST-Mdv1(NT) plus His-Fis1(ΔTM), and GST-Mdv1(NT) plus His-Fis1(ΔTM) and His-Yta4(ΔTM) were used, respectively. Three independent experiments were performed. The top of the column represents the mean (indicated by *ave*), and single group Student's *t* test was carried out to calculate the *p* values. Raw data are available in S1 Data. (TIF)

**S8 Fig. The effect of Dnm1 oligomerization on the GTPase activity of Dnm1 (related to Figs 10 and 11).** GTPase kinetics assays were performed with the indicated proteins: 0.5 μM Dnm1(WT)/Dnm1(G380D) alone or in the presence of 10 μM His-Yta4(ΔTM). The curves were created by fitting to a Michaelis–Menten model, and $K_m$ and $V_{max}$ values were obtained from the fitting. Data points are averages, while error bars represent SD (from 3 independent experiments). Values in the table are average ± SD. Student's *t* test was used to calculate *p* values. Raw data are available in S1 Data. (TIF)

**S1 Raw Images. Unprocessed raw images used in the present work.**
(PDF)

**S1 Data. The individual numerical values used to generate the summary data graphs displayed in the main and supplementary figures.**
(XLSX)

**S1 Movie. (related to Fig 1F).** Mitochondrial dynamics in WT cells. Images were acquired every half a minute with a spinning-disk microscope.
(AVI)

**S2 Movie. (related to Fig 1F).** Mitochondrial dynamics in *yta4Δ* cells. Images were acquired every half a minute with a spinning-disk microscope.
(AVI)

**S3 Movie. (related to Fig 1F).** Mitochondrial dynamics in *yta4Δ* Yta4-13Myc cells. Images were acquired every half a minute with a spinning-disk microscope.
(AVI)

**S4 Movie. (related to Fig 2A).** Mitochondrial dynamics in WT cells treated with 0 μM FCCP. Images were acquired every 1 min.
(AVI)

**S5 Movie. (related to Fig 2A).** Mitochondrial dynamics in WT cells treated with 0.5 μM FCCP. Images were acquired every 1 min.
(AVI)

**S6 Movie. (related to Fig 2A).** Mitochondrial dynamics in *yta4Δ* cells treated with 0 μM FCCP. Images were acquired every 1 min.
(AVI)

**S7 Movie. (related to Fig 2A).** Mitochondrial dynamics in *yta4Δ* cells treated with 0.5 μM FCCP. Images were acquired every 1 min.
(AVI)

**S1 Table. Plasmids used in the paper.**
(DOCX)

**S2 Table. Yeast strains used in the paper.**
(DOCX)

## Acknowledgments

We thank Prof. Quan Chen (Nankai University) and the members in the Fu laboratory for insightful discussion and Dr. Li-lin Du (National Institute of Biological Sciences, China) for providing yeast strains.

## Author Contributions

**Conceptualization:** Chuanhai Fu.

**Formal analysis:** Jiajia He, Ke Liu.

**Funding acquisition:** Xuebiao Yao, Ai-Hui Tang, Chuanhai Fu.

**Investigation:** Jiajia He, Ke Liu, Yifan Wu, Chenhui Zhao, Shuaijie Yan, Jia-Hui Chen, Lizhu Hu, Dongmei Wang, Fan Zheng, Wenfan Wei.

**Methodology:** Jiajia He, Ke Liu, Chao Xu, Chengdong Huang, Xing Liu, Lijun Ding.

**Supervision:** Zhiyou Fang, Ai-Hui Tang, Chuanhai Fu.

**Writing – original draft:** Jiajia He, Zhiyou Fang, Ai-Hui Tang, Chuanhai Fu.

**Writing – review & editing:** Jiajia He, Ke Liu, Chao Xu, Chengdong Huang, Xing Liu, Xuebiao Yao, Lijun Ding, Ai-Hui Tang, Chuanhai Fu.

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
