## [Editor Report · Decision Letter 0]

8 Mar 2022

Dear Dr Fu, 

Thank you for submitting your manuscript entitled "The AAA-ATPase Yta4/ATAD1 interacts with Dnm1 and Fis1 to inhibit mitochondrial fission" for consideration as a Research Article by PLOS Biology.

Your manuscript has now been evaluated by the PLOS Biology editorial staff as well as by an academic editor with relevant expertise and I am writing to let you know that we would like to send your submission out for external peer review.

Once your full submission is complete, your paper will undergo a series of checks in preparation for peer review. Once your manuscript has passed the checks it will be sent out for review. To provide the metadata for your submission, please Login to Editorial Manager (https://www.editorialmanager.com/pbiology) within two working days, i.e. by Mar 10 2022 11:59PM.

If your manuscript has been previously reviewed at another journal, PLOS Biology is willing to work with those reviews in order to avoid re-starting the process. Submission of the previous reviews is entirely optional and our ability to use them effectively will depend on the willingness of the previous journal to confirm the content of the reports and share the reviewer identities. Please note that we reserve the right to invite additional reviewers if we consider that additional/independent reviewers are needed, although we aim to avoid this as far as possible. In our experience, working with previous reviews does save time. 

If you would like to send previous reviewer reports to us, please email me at ialvarez-garcia@plos.org to let me know, including the name of the previous journal and the manuscript ID the study was given, as well as attaching a point-by-point response to reviewers that details how you have or plan to address the reviewers' concerns. 

Given the disruptions resulting from the ongoing COVID-19 pandemic, please expect some delays in the editorial process. We apologise in advance for any inconvenience caused and will do our best to minimize impact as far as possible.

Kind regards,

Ines

--

Ines Alvarez-Garcia, PhD

Senior Editor

PLOS Biology

---

## [Decision Letter · Decision Letter 1]

16 May 2022

Dear Dr Fu,

Thank you for your patience while your manuscript entitled "The AAA-ATPase Yta4/ATAD1 interacts with Dnm1 and Fis1 to inhibit mitochondrial fission" was peer-reviewed at PLOS Biology. Please also accept my sincere apologies for the delay in providing you with our decision. The manuscript has now been evaluated by the PLOS Biology editors, an Academic Editor with relevant expertise, and by two independent reviewers. 

The reviews are attached below. As you will see, the reviewers are generally positive and find the conclusions interesting and novel, however they also think that the results are a bit preliminary and raise several concerns, suggesting a substantial number of experiments that are needed to strengthen the results. Reviewer 1 thinks that the mechanism underlying the regulation of mitochondria fission by Yta4 is confusing and that the final model needs to be fleshed out. Reviewer 2 would like you to analyse the function of Yta4 and its mutants at normal expression levels, rather than by overexpression, and is not convinced about the effects of Yta4 on Dnm1, suggesting several experiments to test if they are functional, among other experiments.

Based on the specific comments of the reviewers and following discussion with the Academic Editor, it is clear that a substantial amount of work would be required to meet the criteria for publication in PLOS Biology. However, given our and the reviewers interest in your study, we would be open to inviting a comprehensive revision of the study that thoroughly addresses all the reviewers' comments and, in general, to strengthen the mechanism by which Yta4 acts on Dnm1 and Fis1 to inhibit mitochondrial fission. Considering the extent of revision that would be needed, we cannot make a decision about publication until we have seen the revised manuscript and your response to the reviewers' comments. Your revised manuscript would need to be seen by the reviewers again, but please note that we would not engage them unless their main concerns have been addressed.

We appreciate that these requests represent a great deal of extra work, and we are willing to relax our standard revision time to allow you 6 months to revise your study. Please email us (plosbiology@plos.org) if you have any questions or concerns, or envision needing an extension.

**IMPORTANT - SUBMITTING YOUR REVISION**

3. Resubmission Checklist

a) *PLOS Data Policy*

b) *Published Peer Review*

d) *Blurb*

Please also provide a blurb which (if accepted) will be included in our weekly and monthly Electronic Table of Contents, sent out to readers of PLOS Biology, and may be used to promote your article in social media. The blurb should be about 30-40 words long and is subject to editorial changes. It should, without exaggeration, entice people to read your manuscript. It should not be redundant with the title and should not contain acronyms or abbreviations. For examples, view our author guidelines: https://journals.plos.org/plosbiology/s/revising-your-manuscript#loc-blurb

Sincerely,

Ines

--

Ines Alvarez-Garcia, PhD

Senior Editor

PLOS Biology

Reviewers' comments

Rev. 1:

In this study, the authors used fission yeast as a model organism to study the role of a conserved mitochondrial outer membrane protein Yta4 (Msp1 in budding yeast and ATAD1 in mammals) in regulating mitochondrial dynamics. Although previous observations by another group showed that Msp1 deletion alone in budding yeast does not cause significant mitochondrial morphology change, the authors showed that deletion of Yta4 in fission yeast causes significant mitochondrial fragmentation, similar to the effect caused by the deletion of ATAD1 in mammals (shown by another group). The authors then showed that the fragmentation takes place in a Dnm1-dependent manner. In the second half of this manuscript, the authors used a combination of imaging and immunoprecipitation assays to establish the functional interactions between Yta4, Dnm1 and Fis1 and reached a model where Yta4 interactions with both Dnm1 and Fis1 and such interactions inhibit excessive mitochondrial fission.

The authors made several key observations/conclusions in this paper. The observation that Yta4 regulates mitochondria dynamics by inhibiting excessive fission is exciting and well supported by experimental data. It points out that Yta4 as an evolutionarily conserved protein, has a role that was previously not well understood, and this role could be conserved in mammals, given that ATAD1 deletion also causes mitochondrial fragmentation. While this observation is exciting, the mechanism by which Yta4 regulates mitochondria fission is still confusing, especially the functional relationship between three key components: Yta4, Dnm1 and Fis1. The manuscript needs more experiments to strengthen the mechanism part. Also, the final model needs to be more clearly spelled out. The current model figure is not informative and needs a lot of improvement. Below are my detailed comments:

Major comments:

My main comments are on Figures 4, 5 and 6, where the authors tried to establish the functional relationships between Yta4, Dnm1 and Fis1.

(1) In Figure 4, the authors showed that overexpression of either a catalytic-dead mutant (Yta4(EQ)) or a translocation-defective mutant (Yta(AA)) of Yta4 causes dislocation of Dnm1 from the mitochondria without dislocalizing Fis1. Given that Dnm1 is recruited to the mitochondria by Fis1, this result suggests that Yta4 could destabilize the interaction between Dnm1 and Fis1, and this destabilization is not dependent on the enzymatic activity or the translocase activity of Yta4. The authors later speculated in the discussion that this implies that Yta4 could dislocate Dnm1 through a direct interaction with Dnm1, which is confusing, because if Yta4(EQ) is able to interact with Dnm1, it should be able to help Dnm1 localize to the mitochondria. So, taking the results of Figure 4, perhaps an explanation to this result that would make more sense is that Yta4's effects of the Dnm1-Fis1 complex is twofold: first, it removes Fis1 from the mitochondria, keeping the homeostatic balance of Fis1 on the outer membrane, and this function is dependent on the translocase activity. Second, its interaction with Fis1 inhibits Dnm1 from interacting with Fis1, and this interaction does not require a translocase or ATPase activity, potentially because monomeric Yta4 could interact with Fis1. So, when the wild-type Yta4 is overexpressed, both Yta4's functions take place, resulting in the dislocation of both Fis1 and Dnm1 from the mitochondria. However, when a mutant version of Yta4 is expressed, it could no longer dislocase Fis1, but still able to inhibit Dnm1-Fis1 interaction, causing the observed dislocation of Dnm1, but not Fis1.

(2) In Figure 5, both in vivo and in vitro pull-downs were performed to examine the physical interaction between Yta4 and Dnm1/Fis1. As the nucleotide-bound state is key to a AAA protein's oligomeric state and function, the authors should do these experiments with nonhydrolyzable nucleotides added and compare to the conditions without nucleotides, as this will help examine whether hexamer formation will enhance these interactions, especially for the one between Yta4 and Fis1. Since the authors' model is that Yta4 helps to destabilize the Fis1-Dnm1 complex, an in vitro assay using the same setup described in this figure should be done to test whether that is the case (pre-immobilize Fis1 to beads and bind Dnm1 to it, and then add Yta4 and ATP to watch dissociation of the Dnm1), and whether that is dependent on the presence of ATP.

(3) In Figure 6B, the authors examined the effect of Yta4 binding to the GTPase activity of Dnm1. Could the authors specify if Yta4 binding changes the Vmax of the reaction? If not, Yta4 binds in a competitive mode, without changing the inherent enzymatic capacity of Dnm1. It would be helpful to distinguish mechanistically (with the data the authors already have) whether Yta4 is a competitive or noncompetitive inhibitor.

(4) Figure 6C suggests that Yta4 can compete out GDP or GTP from binding to Dnm1, but it fails to compete out GMPPNP. There is no explanation to this result. Is that because GMPPNP is known to bind Dnm1 more strongly? It's confusing what the GMPPNP group shows here. Also, it would help to have a group without any nucleotide, as a negative control.

(5) The model figure (Figure 7) needs a lot of improvement. Currently, it is extremely confusing what it is trying to convey. A good way to make this figure could be describing the normal versus the Yta4 deletion scenarios separately, and spell out clearly which component interacts with which component and which interaction helps dislocate which component etc.

(6) The authors mentioned that Dnm1 is recruited to the mitochondria by Fis1, through Cav4/Mdv1. However, there is no examination of Yta4's effect on Cav4/Mdv1's mitochondrial localization throughout this study. To strengthen the mechanistic part of this study, the authors should examine the effect of Yta4 deletion and overexpression on Cav4/Mdv1, and possibly test their physical interactions using the already established in vitro assay, to further dissect the role of Yta4 in the stabilization/destabilization of this big complex and how that leads to the observed morphological change in the organelle.

Minor comments:

(1) Msp1 is misspelled as Mps1 three times. Twice in page 3 and once in page 6.

(2) The bottom panel of Figure 3A showed mitochondrial constriction (but not fission) in the absence of Dnm1. Can the authors comment on the cause of that?

(3) Figure 1B: I assume the top-right panel refers to mitochondria per cell, whereas the bottom panels refer to junctions per mitochondria (not per cell)? If so, the authors should specify this in the text.

(4) Figure 1G: should the bottom panel be "fusion" instead of "fission"?

(5) There are a few places where the authors observed clumped mitochondria. For example, in Figure 4D, expression of Yta4 (EQ) leads to clumping. Is that a sign of inhibited fusion? The authors should comment on the cause of clumping, and why that is observed only when the dead mutant is expressed by not in the Yta4 deletion cells.

(6) Figure 5D and 5F: in the GST blot, what is the top band in the third lane? It is only present in the control lane but not in the experiment lane.

Rev. 2:

Msp1/ATAD1 in budding yeast and mammals as an AAA-ATPase in the mitochondrial outer membrane (OM) mediates the quality control of mistargeted tail-anchor (TA) protein on the mitochondrial surface; Msp1 extracts mistargeted TA proteins from the mitochondrial OM and transfers them to the ER, where mistargeted proteins are subjected to the ER protein quality control. In the present study, He et al. report a new possible role of Yta4, an Msp1 homolog in fission yeast, in the control of the mitochondrial morphology. The observation is interesting and may attract the interest of the broad readership of the journal. However, the manuscript has many points to be improved.

Major points.

1) It is of interest if the observed effects of Yta4 on Dnm1 functions reflect the possible ability of Yta4 to extract Fis1 (and Dnm1?) from the mitochondrial OM. That is, does Yta4 indeed extract a fraction of normal, not mistargeted Fis1, from the OM to suppress the level of recruited Dnm1? This can be tested by following the fate of Fis1 upon expression of Yta4 in yta1� cells. In relation to this, it is strange that, while the level of GFP-Fis1 increased with overexpressed WT Yta4 (Fig. 4C), the GFP-Fis1 signal decreased under the microscope (Fig. 4E). If Fis1 was extracted from the OM but not degraded, is the extracted Fis1 transferred to the ER?

2) Since overexpression of Yta4 could cause, like in the case of Msp1, secondary effects through perturbed TA protein trafficking, functions of Yta4 and its mutants should be analyzed at their normal expression levels by using the YTA4 own promoter (Fig. 4). Co-IP experiments should be performed with cells with normal levels of Yta4 or WT cells, not the cells with overexpressed Yta4 in Fig. 5C.

3) In Fig. 6, the authors tested the effects of Yta4 on the GTPase activity of Dnm1 by using recombinant Dnm1 and Yta4. However, these experiments are problematic, and the results on the effects of Yta1 on Dnm1 functions appear preliminary.

- It is not clear if Dnm1 and Yta4 are functional; they only tested the GTPase activity of Dnm1. Does Yta4 show the expected ATPase activity? Does purified Yta4 exist as a monomer or oligomer?

- The effects of Yta4 on possible filament formation of Dnm1 were tested by centrifugation in Fig. 6C. However, it is not clear if Dnm1 in the pellet reflects filament formation or just aggregate formation.

- It is unclear why they tested only single concentrations of the proteins: Dnm1 = 0.5 uM and Yta4 =5 uM.

- The nucleotide-bound state of Dnm1 should affect the localization of Dnm1 to mitochondria as well. This should be considered.

- GTP may be hydrolyzed to GDP under the condition of Fig. 6C.

4) Fig. 4D suggests that Fis1 interacting with the Yta4 mutants cannot recruit Dnm1 to mitochondria. This should be directly tested.

Minor points

Page 3 - Fmp32 (Put6) localization needs a reference of https://www.nature.com/articles/s41467-020-18704-1

Ftr1 should read Frt1, and Yst6 should read Ysy6.

Fig. 4D and E - Why mitochondria become aggregated with overexpressed mutant Yta1?

Fig. 4B and text - Kcat should read kcat.

Fig. 4E - The signals from Mito Tracker are too strong and saturated.

Fig. 5A - Yta4-tdTomato as a homolog of Msp1 is expected to localize not only to mitochondria but also peroxisomes. However, peroxisomal localization of Yta4 is not obvious here. Why?

Fig. 5C - The positions of Yta4-13myc and GFP-Fis1 are not the same among the three gels for each. If the rightmost panel is to show that GFP IP can pull down Dnm1, it needs control without GFP-Fis1 expression.

Fig. 5D and E - Input controls are missing.

---

## [Decision Letter · Decision Letter 2]

12 May 2023

Dear Dr Fu,

Thank you for your patience while we considered your revised manuscript entitled "The AAA-ATPase Yta4/ATAD1 interacts with the mitochondrial divisome to inhibit mitochondrial fission" for consideration as a Research Article at PLOS Biology. Your revised study has now been evaluated by the PLOS Biology editors, the Academic Editor and two of the original reviewers 

The reviews are attached below. You can see that both reviewers appreciate the improvements in the manuscript and Reviewer 2 is now mostly satisfied. Reviewer 1, however, raises some remaining points that would need to be addressed. These require to flesh out the role of Dnm1 oligomerization in yta4’s inhibition to its activity, performing a quantification and clarifying some points in the model of Yta4’s regulation on mitochondrial fission.

In light of the reviews, which you will find at the end of this email, we are pleased to offer you the opportunity to address the remaining points from the reviewers in a revision that we anticipate should not take you very long. We will then assess your revised manuscript and your response to the reviewers' comments with our Academic Editor aiming to avoid further rounds of peer-review, although might need to consult with the reviewers, depending on the nature of the revisions.

**IMPORTANT - SUBMITTING YOUR REVISION**

3. Resubmission Checklist

a) *PLOS Data Policy*

b) *Published Peer Review*

Sincerely,

Ines

--

Ines Alvarez-Garcia, PhD

Senior Editor

PLOS Biology

Reviewers' comments

Rev. 1:

The authors made significant improvements on the previous manuscript. Now, the model of Yta4’s regulation on mitochondrial fission consists of three parts: (1) Yta4 extracts Fis1 from the OMM. (2) Yta4 inhibits the interaction between Mdv1 and Dnm1 (3) Yta4 is a competitive inhibitor of the GTPase activity of Dnm1.

Despite the experiments added, I still have concerns over some of these claims.

For the first claim, I am convinced that Fis1 is a Yta4 substrated when Yta4 is overexpressed. However, the authors showed that under normal Yta4 expression level, Fis1’s localization is on the mitochondria, and not significant changes in localization was observed when Yta4 is deleted. To claim that Yta4 removes Fis1 from the OMM under regular expression levels, the authors will need to quantify the level of Fis1 on the OMM in WT and KO conditions. Only when there is a change in Fis1’s level on the OMM, can the authors claim that Yta4 removes Fis1 to regulate mito fission.

For the second claim, the authors showed the Yta4 inhibits oligomer formation of Dnm1, since that is presented as a substantial piece of data in the paper, the authors should clarify whether the inhibition of Dnm1-Mdv1 interaction observed is due to the inhibition of oligomerization of Dnm1. That is, whether oligomerization of Dnm1 is required for the Dnm1-Mdv1 interaction.

For the third claim, I am convinced that Yta4 inhibits the GTPase activity. But again, how is this related to the inhibition of oligomerization? Since the authors presented a three-fold mechanism here, they should make sure that the three reasons are orthogonal to each other. The role of Dnm1 oligomerization in yta4’s inhibition to its activity needs to be fleshed out.

Overall, I recommend more experiments and text revision to flesh out the model before acceptance of publication.

Rev. 2:

This is a revised version of the manuscript previously submitted to PLOS Biology. The authors performed new experiments extensively and incorporated significantly new results, which further fortify the model presented in Fig. 12. Although the comparison and interpretation of the in vitro and in vivo situations are still complex, the flow of the logic is more seamless than before. The distinct roles of Msp1/Yta4/ATAD1 in mitochondrial morphology are exceedingly intriguing from an evolutionary point of view. Hence, the current manuscript appears ready for publication.

---

## [Editor Report · Decision Letter 3]

26 Jun 2023

Dear Dr Fu,

Thank you for your patience while we considered your revised manuscript entitled "The AAA-ATPase Yta4/ATAD1 interacts with the mitochondrial divisome to inhibit mitochondrial fission" for publication as a Research Article at PLOS Biology. This revised version of your manuscript has been evaluated by the PLOS Biology editors and the Academic Editor.

Based on our Academic Editor's assessment of your revision, we are likely to accept this manuscript for publication, provided you satisfactorily address the data and other policy-related requests stated below.

In addition, the Academic Editor would like you to re-phrase the sentence stating: "In evolution, the fission yeast Schizosaccharomyces pombe is more ancient than the budding yeast Saccharomyces cerevisiae" to "The fission yeast Schizosaccharomyces pombe exhibits more ancient traits than the budding yeast Saccharomyces cerevisiae."

We expect to receive your revised manuscript within two weeks. 

*Published Peer Review History*

*Press*

Sincerely,

Ines

--

Ines Alvarez-Garcia, PhD

Senior Editor

PLOS Biology

Fig 1B, D, G; Fig. 2B; Fig. 3B, E; Fig. 9B, D; Fig, 10B, C; Fig. 11B, D, E; Fig. S3D, E; Fig. S4B, C, E; Fig. S6C; Fig. S7B and Fig. S8

---

## [Editor Report · Decision Letter 4]

12 Jul 2023

Dear Dr Fu,

Thank you for the submission of your revised Research Article entitled "The AAA-ATPase Yta4/ATAD1 interacts with the mitochondrial divisome to inhibit mitochondrial fission" for publication in PLOS Biology. On behalf of my colleagues and the Academic Editor, Sophie Martin, I am delighted to let you know that we can in principle accept your manuscript for publication, provided you address any remaining formatting and reporting issues. These will be detailed in an email you should receive within 2-3 business days from our colleagues in the journal operations team; no action is required from you until then. Please note that we will not be able to formally accept your manuscript and schedule it for publication until you have completed any requested changes.

PRESS

Sincerely, 

Ines

--

Ines Alvarez-Garcia, PhD

Senior Editor

PLOS Biology
